# HSD17B7 is required for the function of sensory hair cells by regulating cholesterol synthesis

Yuqian Shen[1†], Ziyang Wang[1†], Xun Wang[1], Fuping Qian[2], Mingjun Zhong[3], Xin Wang[2]*, Jing Cheng[3]*, Dong Liu[1]*

[1]Co-innovation Center of Neuroregeneration, School of Life Sciences, Nantong Laboratory of Development and Diseases, Nantong, China; [2]Institute of Special Environmental Medicine, Nantong University, Nantong, China; [3]Institute of Rare Diseases, West China Hospital of Sichuan University, Chengdu, China

*For correspondence:
ntuwx@ntu.edu.cn (XW);
chengjing@wchscu.cn (JC);
liudongtom@gmail.com (DL)

[†]These authors contributed equally to this work.

Competing interest: The authors declare that no competing interests exist.

## eLife Assessment

This study provides **valuable** data on the role of Hsd17b7, a gene involved in cholesterol biosynthesis, as a potential regulator of mechanosensory hair cell function. The authors used both zebrafish and the HEI cell line to examine the effects of deletion of Hsd17b7 on hair cell function and survival. While the study presents **convincing** evidence, the effect sizes observed across several experiments, including functional readouts such as the acoustic startle response, are modest, which raises questions about the biological significance of the proposed mechanism.

**Abstract** Cholesterol homeostasis is fundamental to cellular function, and its disruption underlies a wide range of human diseases. However, the contribution of cholesterol biosynthesis to auditory physiology remains poorly understood. HSD17B7 (17β-Hydroxysteroid dehydrogenase type 7) catalyzes the conversion of zymosterone to zymosterol, a key step in the post-lanosterol cholesterol biosynthetic pathway. Here, we found that Hsd17b7 is highly enriched in sensory hair cells of zebrafish and mice. The deficiency of Hsd17b7 reduced intracellular cholesterol levels in HEI-OC1 cells and zebrafish hair cells, thereby compromising MET and acoustic startle responses. A heterozygous nonsense variant (c.544G>T; p.E182*) in *HSD17B7* was identified in an individual with bilateral profound hearing loss. mRNA of c.544G>T HSD17B7 failed to rescue the impaired MET and acoustic startle response of hsd17b7 mutants. Mechanistically, the mutation decreases mRNA abundance and significantly reduces protein. Moreover, expression of the p.E182* mutation disrupted the interaction between HSD17B7 and the ER retention receptor RER1, leading to aberrant subcellular localization and altered cholesterol distribution, thereby exacerbating HC dysfunction. Together, our findings suggest a conserved and essential role for HSD17B7-mediated cholesterol biosynthesis in sensory hair cell function and identify HSD17B7 as a candidate gene for sensorineural hearing loss.

## Introduction

Hearing loss is one of the most common sensory disorders worldwide and represents a major global health burden. According to the World Health Organization, more than 466 million people currently suffer from disabling hearing loss, and this number is expected to increase substantially in the coming decades (*Hu et al., 2016*). Genetic factors account for approximately 60% of congenital hearing loss, and mutations in genes that regulate the development and function of sensory hair cells (HCs) are the main cause of hereditary deafness. Although more than 150 non-syndromic deafness genes have

been identified (*Ingham et al., 2019*), the molecular mechanisms underlying HC dysfunction remain incompletely understood, and many pathogenic genes remain to be discovered.

HCs are the primary mechanoreceptors of the auditory and vestibular systems, converting mechanical stimuli into electrical signals through mechanotransduction (MET). This process critically depends on the integrity and biophysical properties of HC membranes, particularly those of the stereocilia. Cholesterol is the most abundant sterol molecule in mammalian cells and plays essential roles in maintaining membrane structure, synthesizing important hormones, facilitating synapse formation, and mediating cell signaling transduction (*Pfrieger and Ungerer, 2011*; *Yoon et al., 2022*; *Dietschy and Turley, 2001*; *Wu et al., 2023*). Previous studies have shown that dysregulated intracellular cholesterol homeostasis contributes to auditory defects (*Wu et al., 2023*; *Ding et al., 2020*; *Levic and Yamoah, 2011*), including hereditary (*Wang et al., 2019*; *Yao et al., 2019*; *Xing et al., 2015*; *Maas et al., 2017*; *King et al., 2014*), noise-induced (*Wang et al., 2020*; *Sai et al., 2020*; *Milon et al., 2021*), ototoxic (*Zhou et al., 2018*; *Ory et al., 2017*), and age-related hearing loss (*Tang et al., 2019*; *Honkura et al., 2019*; *Sodero et al., 2023*). HSD17B7 is a member of the hydroxysteroid dehydrogenase family and functions as a key enzyme in the cholesterol biosynthetic pathway by catalyzing the conversion of zymosterone to zymosterol. In addition to its role in steroid metabolism (*Saloniemi et al., 2012*) and its involvement in hormone-related cancers (*Shehu et al., 2011*; *Hilborn et al., 2017*), HSD17B7 has been reported to be expressed in multiple tissues, including the brain, eye, and inner ear (IE; *Shehu et al., 2008*; *Marijanovic et al., 2003*). Notably, single-cell RNA sequencing (scRNA-seq) and immunostaining analyses have suggested that HSD17B7 is enriched in mouse vestibular HCs (*Jan et al., 2021*). Despite this expression pattern, the functional role of HSD17B7 in HCs and MET has not been investigated, and no pathogenic variants have been reported to be associated with hearing loss.

Here, our findings reveal a previously unrecognized link between HC–intrinsic cholesterol biosynthesis and the function of sensory HCs, and identify HSD17B7 as a candidate gene underlying sensory hearing disorders.

## Results

### *Hsd17b7 is* expressed in sensory hair cells of zebrafish and mice

To assess the role of Hsd17b7 in the auditory system, we first investigated the expression of *hsd17b7* in the developing zebrafish. scRNA-seq data analysis (accession no. GSE221471; *Qian et al., 2022*) categorized lateral line hair cells (LLHCs; cluster 0), supporting cells (cluster 1), macula hair cells (MHCs; cluster 2), crista hair cells (CHCs 1; cluster 3), mantle cells (cluster 4), and CHCs 2 (cluster 5) using the Seurat 4.0.1 platform (*Figure 1A*). The feature plot and violin plot showed that *hsd17b7* was expressed in sensory HCs, especially in LLHCs and CHCs (*Figure 1B and C*). Quantitative comparison of average expression levels within LLHCs indicates that *hsd17b7* is expressed at a level comparable to several known MET-associated genes (e.g. *tmc1* and *lhfpl5a*; *Figure 1D*). To examine the temporal dynamics of *hsd17b7* expression during LLHCs differentiation, LLHCs, supporting cells, and mantle cells were ordered along pseudotime using Monocle 3 (*Figure 1E*). A heatmap was generated to visualize the expression dynamics of marker genes along the LLHC developmental trajectory (*Figure 1F*). These analyses suggested a gradual increase in *hsd17b7* expression during LLHC maturation (*Figure 1E and F*). Consistently, whole-mount in situ hybridization (WISH) further showed that *hsd17b7* was specifically enriched in neuromast and crista regions at 72 and 96 hours post-fertilization (hpf; *Figure 1G*). To further explore the subcellular localization of *hsd17b7* in zebrafish HCs, we generated a *hsd17b7-egfp* fusion construct driven by a HC-specific *myo6b* promoter (*Maeda et al., 2017*; *Kindt et al., 2012*). Confocal imaging revealed that hsd17b7-EGFP localized predominantly to the cytoplasm in both CHCs and LLHCs, exhibiting a punctate distribution pattern (*Figure 1H and I*).

To further elucidate Hsd17b7 expression in the mammalian auditory system, we analyzed three published scRNA-seq datasets from mouse IE tissues (*Burns et al., 2015*; *Xu et al., 2022*; *Wilkerson et al., 2021*). These analyses revealed that Hsd17b7 was enriched in sensory hair cells, including outer hair cells (OHCs), inner hair cells (IHCs), and utricular hair cells (UHCs), while lower expression levels were observed in CHCs (*Figure 1J*). Next, we performed immunostaining to assess Hsd17b7 protein expression in the organ of Corti. Hsd17b7 was detected in both OHCs and IHCs at P1, P4, and P7 (*Figure 1K*). The specificity of the Hsd17b7 antibody was further supported by independent validation

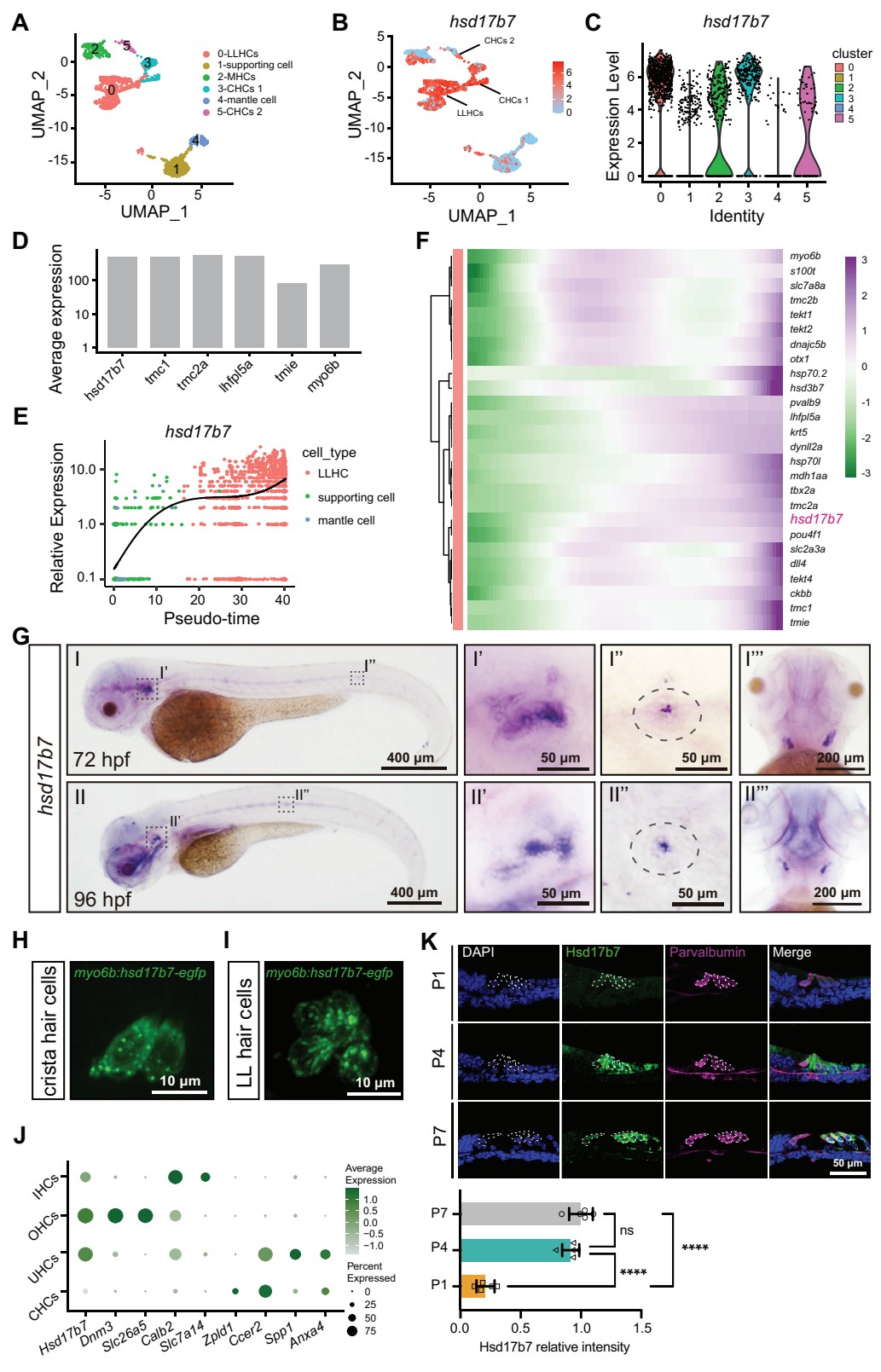

**Figure 1.** Conserved expression of *Hsd17b7* in zebrafish and mice sensory hair cells. (**A**) The UMAP analysis of the zebrafish scRNA-seq data. LLHCs, lateral line hair cells; MHCs, macula hair cells; CHCs, crista hair cells; (**B, C**) Feature plot and violin plot of *hsd17b7* expression across cell types. (**D**) Quantitative comparison of average expression levels of hsd17b7 within LLHCs. (**E**) *hsd17b7* expression increased along the pseudotime trajectory

*Figure 1 continued on next page*

*Figure 1 continued*

of LLHC formation. (**F**) Heatmap of marker genes along the pseudotime trajectory. (**G**) The expression of the *hsd17b7* in 72 hpf and 96 hpf embryos was detected by whole-mount in situ hybridization analysis. Dashed circles indicate the otic vesicle and the neuromast. (**H, I**) Representative images of the CHCs and LLHCs in *Tg(myo6b: hsd17b7-egfp)* at 4 dpf. (**J**) The average expression of Hsd17b7 and HC marker genes across mouse HC subtypes from published scRNA-seq datasets. UHCs, utricular hair cells. (**K**) Immunostaining and quantification analysis of HSD17B7 in dissected mouse organ of Corti sections at P1, P4, and P7. Dashed circles indicate outer hair cells (OHCs) and inner hair cells (IHCs).

The online version of this article includes the following source data and figure supplement(s) for figure 1:

**Figure supplement 1.** Validation of HSD17B7 antibody specificity in HEI-OC1 cells.

**Figure supplement 1—source data 1.** Original files for western blot analysis displayed in *Figure 1—figure supplement 1B*.

**Figure supplement 1—source data 2.** Original western blot analysis displayed in *Figure 1—figure supplement 1B*, labelled.

**Figure supplement 2.** Evolutionary conservation of Hsd17b7.

---

experiments in HEI-OC1 cells (*Figure 1—figure supplement 1*). These results indicate that Hsd17b7 is expressed and enriched in HCs in both zebrafish and mice.

In addition, phylogenetic and sequence alignment analysis revealed that vertebrate HSD17B7s share a significant similarity in amino acid sequences (*Figure 1—figure supplement 2A*). UniProt sequence alignment analysis revealed that *Danio rerio* hsd17b7 (NP_001070796.1) shares 58.2% identity with *Homo sapiens* HSD17B7 (NP_057455.1) and 67.1% identity with *Mus musculus* Hsd17b7 (NP_034606.3), respectively (*Figure 1—figure supplement 2B*). These data suggest that Hsd17b7 is conserved and expressed in HCs across vertebrate species.

## Hsd17b7 deficiency impaired acoustic startle responses and MET activity in zebrafish

To investigate the *hsd17b7* function in the auditory system, we generated *hsd17b7* mutant alleles using CRISPR/Cas9-mediated genome editing (*Figure 2—figure supplement 1A–D*). Sequence analysis identified multiple mutant alleles, including a 4 bp deletion and two 6 bp deletions in exon 3 (*Figure 2—figure supplement 1E*). The 4 bp deletion allele was selected for further analysis, as it led to a frameshift and premature stop codon, producing a severely truncated protein lacking most of the conserved catalytic domain required for *hsd17b7* enzymatic activity (*Figure 2—figure supplement 1F*).

We first examined acoustic startle responses (*Yang et al., 2017*) at 5 dpf (*Figure 2A*). Compared with control larvae, *hsd17b7* mutants exhibited significantly reduced movement trajectories, peak swimming velocity, and total travel distance following acoustic stimulation (*Figure 2A–C*), indicating impaired startle response. Microinjection of *hsd17b7* mRNA into mutant embryos significantly restored these behavioral phenotypes (*Figure 2A–C*), confirming that the phenotype was specifically caused by loss of Hsd17b7.

To validate these findings, antisense morpholino oligonucleotides (*hsd17b7* Mo) targeting the junction of exon 1/intron 1 were designed to knock down hsd17b7 expression (*Figure 2—figure supplement 2A and B*). RT-PCR and immunoblotting analyses confirmed effective disruption of normal splicing and a substantial reduction in Hsd17b7 protein levels, both of which were rescued by co-injection of *hsd17b7* mRNA (*Figure 2—figure supplement 2C and D*). Consistent with the mutant phenotype, *hsd17b7* morphants displayed significantly reduced acoustic startle responses at 5 dpf, including decreased movement trajectories, swimming distance, and peak velocity, all of which were effectively rescued by *hsd17b7* mRNA injection (*Figure 2—figure supplement 2E–G*).

In the *Tg(Brn3c: mGFP)* background (*Qian et al., 2022*; *Xiao et al., 2005*), confocal imaging revealed a significant reduction in the number of LLHCs in *hsd17b7* mutants (~16% reduction; *Figure 2D and E*). To assess whether *hsd17b7* loss affects the MET activity, we performed FM4-64 uptake assays, which label HCs through functional MET channels (*Marijanovic et al., 2003*; *Xu et al., 2022*; *Wilkerson et al., 2021*). Quantification of FM4-64 fluorescence intensity per HC revealed a pronounced reduction in MET-dependent dye uptake in *hsd17b7* mutant HCs at 5 dpf (~40%; *Figure 2F and G*).

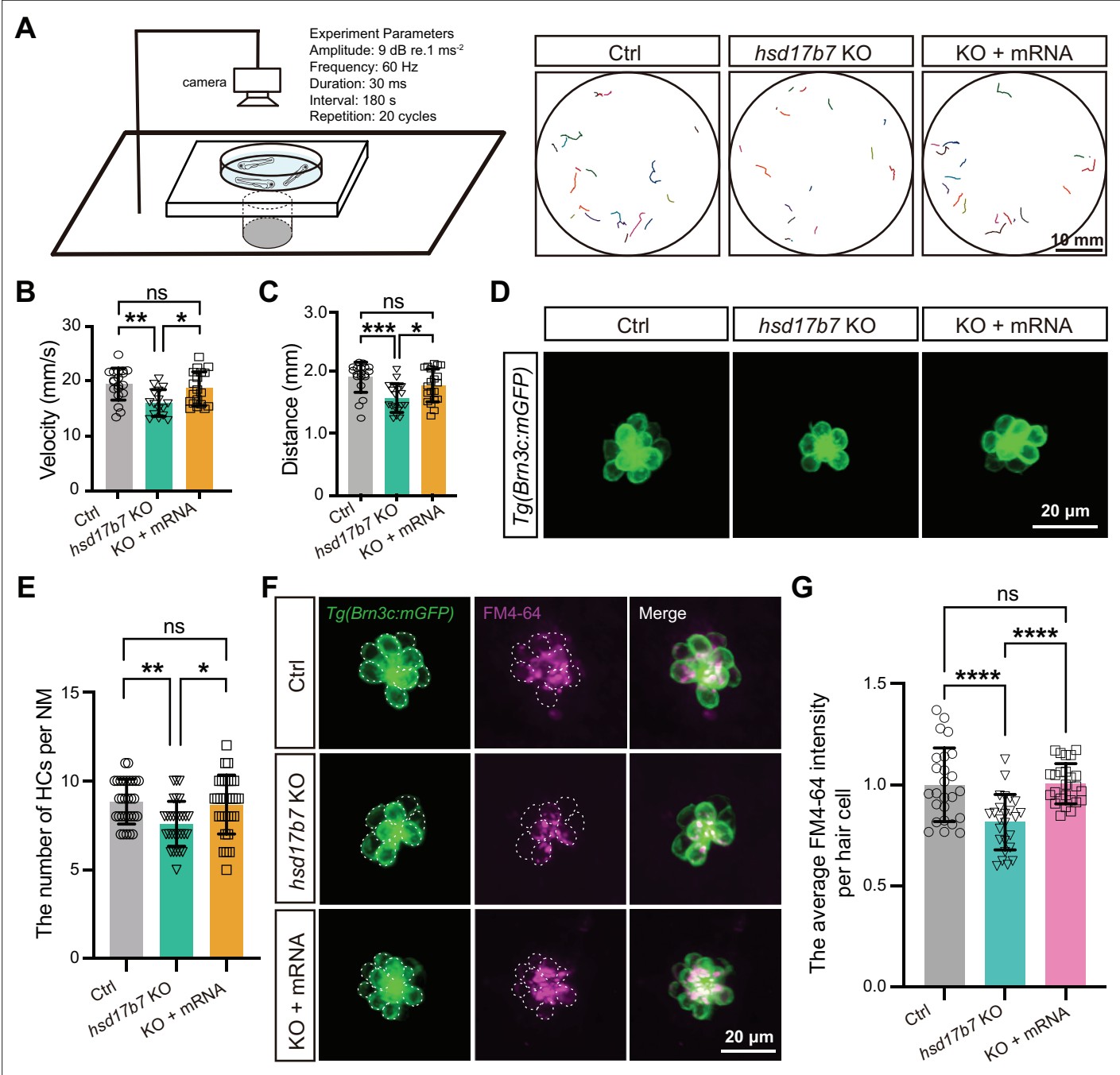

**Figure 2.** Loss of hsd17b7 impaired acoustic startle responses and mechanotransduction activity in zebrafish. (**A**) Left, schematic of the experimental setup used to assess acoustic startle responses in zebrafish larvae. Right, representative locomotor trajectories showing C-start responses to a single acoustic stimulus (9 dB re. 1 m·s⁻², 60 Hz tone burst) in control, *hsd17b7* knockout (KO), and *hsd17b7* mRNA–rescued larvae at 5 dpf. Scale bars, 10 mm. (**B, C**) Quantification of peak swimming velocity (**B**) and total movement distance (**C**) in response to acoustic stimulation shown in (**A**) (n=20). One-way ANOVA followed by Tukey's multiple comparisons, *p<0.05, **p<0.01, ***p<0.001, ns, non-significant (p>0.05), mean ± SEM. (**D**) Representative confocal images of lateral line hair cells (LLHCs, green) in *Tg(Brn3c:mGFP)* larvae at 4 dpf from control, *hsd17b7* KO, and *hsd17b7* mRNA–rescued groups. Scale bars, 20 μm. (**E**) Quantification of HC number per neuromast (n=30). One-way ANOVA followed by Tukey's multiple comparisons, *p<0.05, **p<0.01, ns, non-significant (p>0.05), mean ± SD. (**F**) Representative images of LLHCs (green) and FM4-64-labeled functional HCs (magenta) in single neuromasts of Tg(Brn3c:mGFP) larvae at 5 dpf from the indicated groups. White dashed outlines indicate LLHCs. Scale bars, 20 μm. (**G**) Quantification of relative FM4-64 fluorescence intensity per neuromast (n=25). One-way ANOVA followed by Tukey's multiple comparisons, ****p<0.0001, ns, non-significant (p>0.05), mean ± SD.

The online version of this article includes the following source data and figure supplement(s) for figure 2:

*Figure 2 continued on next page*

*Figure 2 continued*

**Figure supplement 1.** Generation of *hsd17b7* mutant using the CRISPR/Cas9 system in zebrafish.

**Figure supplement 2.** Knockdown of *hsd17b7* impairs auditory-evoked startle responses in zebrafish.

**Figure supplement 2—source data 1.** Original files for nucleic acid gel electrophoresis analysis displayed in *Figure 2—figure supplement 2C*.

**Figure supplement 2—source data 2.** Original nucleic acid gel electrophoresis analysis displayed in *Figure 2—figure supplement 2C*, labelled.

**Figure supplement 2—source data 3.** Original files for western blot analysis displayed in *Figure 2—figure supplement 2D*.

**Figure supplement 2—source data 4.** Original western blot analysis displayed in *Figure 2—figure supplement 2D*, labelled.

**Figure supplement 3.** Knockdown of *hsd17b7* decreases FM4-64 uptake in hair cells.

A similar decrease was observed in *hsd17b7* morphants (~25%; *Figure 2—figure supplement 3*). Importantly, *hsd17b7* mRNA injection restored both FM4-64 uptake and HC number in mutants and morphants (*Figure 2D-G*, *Figure 2—figure supplement 3*), indicating that *hsd17b7* is required for normal MET activity in HCs.

## Hsd17b7 deficiency disrupted cholesterol-associated transcriptional programs in hair cells

To investigate intrinsic transcriptional changes in HCs caused by hsd17b7 deficiency, we performed scRNA-seq on FACS-isolated mGFP+ HCs from control and *hsd17b7* mutant (*Figure 3A*). Clustering analysis identified major IE and lateral line (LL) cell populations, including HCs and SCs (*Figure 3B*; *Figure 3—figure supplement 1*). Focusing on LLHCs, pseudobulk differential expression analysis revealed extensive transcriptional remodeling in *hsd17b7* mutants, with 1355 genes differentially expressed, including 853 upregulated (63%) and 502 downregulated (37%) genes relative to controls (*Figure 3C*). Gene Ontology (GO) enrichment analysis of differentially expressed genes highlighted pathways related to lipid metabolism, membrane organization, and sensory perception, indicating broad alterations in cellular homeostasis upon disruption of cholesterol biosynthesis (*Figure 3D and E*).

To further assess cholesterol- and MET-associated transcriptional programs, we performed module score analysis in both LLHCs and IEHCs. In mutant IEHCs, MET, cholesterol biosynthesis, and cholesterol uptake-related gene sets were significantly upregulated, whereas gene modules associated with tip-link components and cholesterol efflux were concurrently reduced (*Figure 3F and G*).

In mutant LLHCs, module score analysis revealed a partially overlapping transcriptional response. Gene modules associated with tip-link components and cholesterol efflux were reduced, whereas modules representing the core MET machinery and cholesterol biosynthesis enzymes did not show significant global changes, consistent with selective remodeling rather than uniform pathway activation or repression. At the gene level, cholesterol regulatory programs appeared partially uncoupled, with downregulation of sterol-sensing transcriptional regulators (e.g. *srebf2, insig1,* and *mbtps1*) accompanied by variable expression changes among cholesterol biosynthetic enzymes. In line with this pattern, gene set enrichment analysis (GSEA) indicated an overall attenuation of cholesterol biosynthesis–associated pathways in mutant LLHCs (*Figure 3H*).

Together, these findings indicate that *hsd17b7* deficiency induces selective and context-dependent remodeling of cholesterol- and MET-associated transcriptional programs across HC populations.

## Hsd17b7 is required for maintaining cholesterol levels in hair cells

Hsd17b7 converts zymosterone to zymosterol, participating in cholesterol biosynthesis (*Figure 4A*; *Marijanovic et al., 2003*). Given that both elevated and reduced cholesterol levels are detrimental to auditory function (*Ding et al., 2020*; *Crumling et al., 2012*), we first verified whether hsd17b7 regulates cholesterol homeostasis in HCs.

In vitro, knockdown of *Hsd17b7* in HEI-OC1 cells resulted in a significant reduction in total cellular cholesterol levels (*Figure 4B and C*). To further examine cholesterol distribution, we employed the D4H cholesterol-binding probe derived from perfringolysin O, which selectively labels cholesterol at the cytoplasmic leaflet of the plasma membrane (*Gao et al., 2022*; *Schoop et al., 2021*). Overexpression of human *HSD17B7* markedly increased D4H-mCherry fluorescence intensity in HEI-OC1

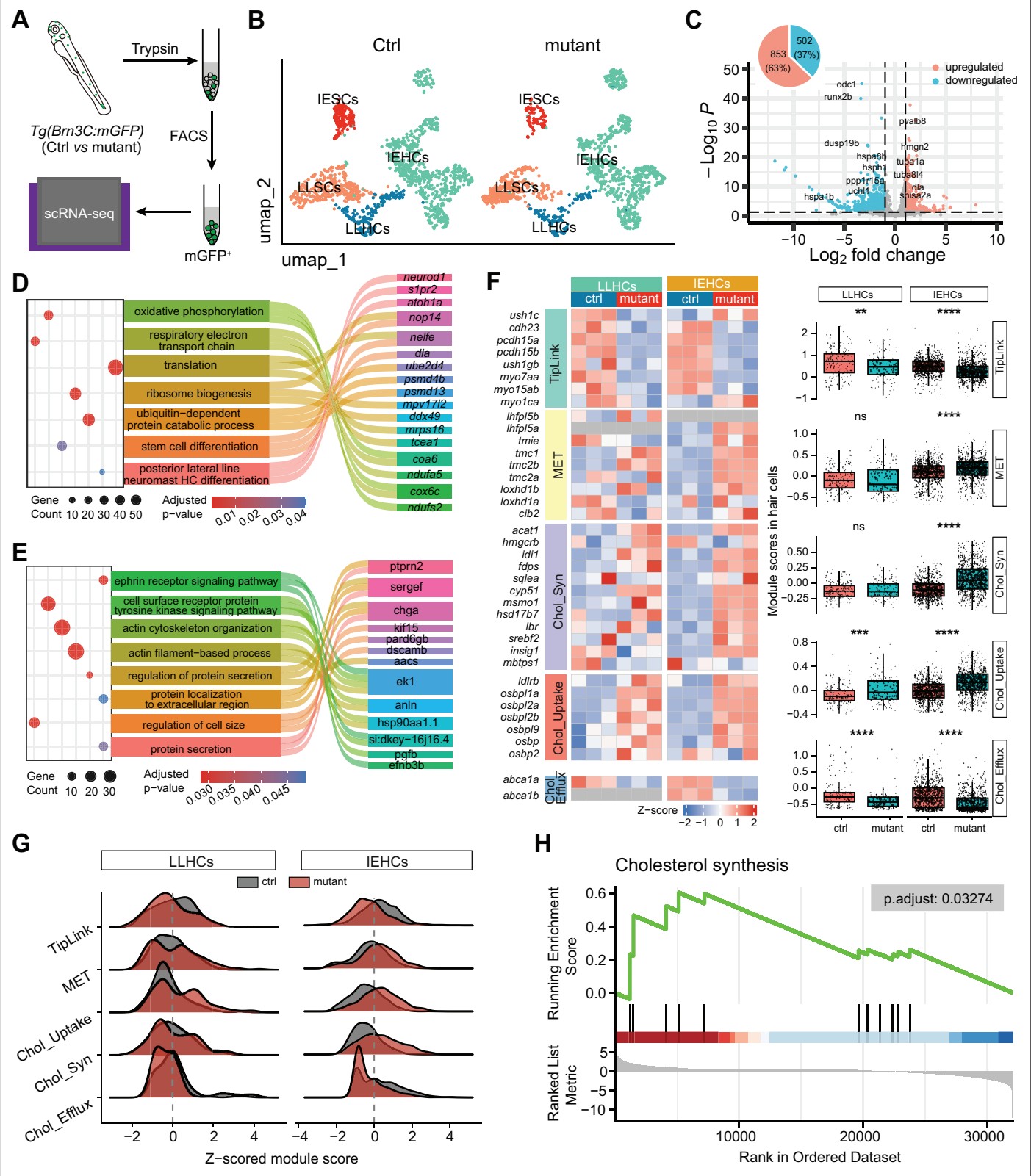

**Figure 3.** Hsd17b7 deficiency disrupted cholesterol-associated transcriptional states in hair cells. (**A**) Schematic overview of the scRNA-seq workflow. mGFP+ hair cells were isolated from control and hsd17b7 mutant zebrafish by fluorescence-activated cell sorting (FACS) and subjected to 10× Genomics single-cell RNA sequencing. (**B**) UMAP visualization of scRNA-seq data from control and hsd17b7 mutant samples, colored by annotated cell types, including inner ear (IE) and lateral line (LL) hair cells (HCs) and supporting cells (SCs). (**C**) Volcano plot showing pseudobulk differential gene

*Figure 3 continued*

expression analysis of LLHCs from hsd17b7 mutants relative to controls. Genes with adjusted p<0.05 are highlighted, with upregulated genes shown in red and downregulated genes shown in blue. (**D and E**) Gene Ontology (GO) biological process enrichment analysis of genes upregulated and downregulated in hsd17b7 mutant HCs. A Sankey diagram illustrates representative genes contributing to each enriched GO term. (**F**) Heatmap and module score analysis of gene sets related to tip-link structure, mechanotransduction (MET), and cholesterol metabolism in LLHCs and IEHCs. Module scores were calculated using predefined gene sets, and statistical significance was assessed using the Wilcoxon rank-sum test. (**G**) Ridge plots showing the distribution of z-scored module scores associated with tip link structure, cholesterol uptake, and cholesterol efflux in control and hsd17b7 mutant LLHCs and IEHCs, revealing population-wide transcriptional shifts rather than discrete subpopulation changes. (**H**) Gene set enrichment analysis (GSEA) demonstrating significant downregulation of cholesterol biosynthesis–associated pathways in hsd17b7 mutant LLHCs.

The online version of this article includes the following figure supplement(s) for figure 3:

**Figure supplement 1.** Validation of cell type annotation and cholesterol- and MET-related transcriptional states in hair cells.

cells compared with controls (*Figure 4D and E*), indicating that HSD17B7 positively regulates cellular cholesterol abundance.

To further assess whether *hsd17b7* regulates cholesterol in vivo, we expressed D4H-mCherry in control and *hsd17b7* mutant zebrafish HCs using the *Tg(Brn3c:mGFP)* background (*Figure 4F*). Consistent with previous reports (*Gao et al., 2022*), D4H-mCherry was enriched in the hair bundle (*Figure 4G*). Notably, cholesterol levels, as indicated by D4H-mCherry intensity, were significantly reduced in both CHCs and LLHCs of *hsd17b7* mutants at 4 dpf (*Figure 4G and H*). A comparable reduction was observed in *hsd17b7* morphants (*Figure 4—figure supplement 1*).

Together, these results demonstrate that *hsd17b7* is required for maintaining cholesterol levels in HCs, both in vitro and in vivo. Given the essential role of cholesterol homeostasis in MET and auditory function (*Ding et al., 2020; Crumling et al., 2012; Cheng et al., 2025*), loss of Hsd17b7 likely compromises MET activity and acoustic startle responses by disrupting cholesterol availability in HCs.

## Identification of a candidate HSD17B7 nonsense variant associated with hearing loss

Given the established role of Hsd17b7 in HCs, we investigated whether variants in the human *HSD17B7* might be associated with hearing loss. Whole-genome sequencing (WGS) was performed in individuals with hearing loss who remained undiagnosed after screening of 201 known deafness-associated genes (*Cheng et al., 2025*). We identified a heterozygous nonsense variant in *HSD17B7* (NM_016371.4), c.544G>T (p.E182*), in a child with bilateral profound congenital hearing loss (*Figure 5A–C*). The proband failed the newborn hearing screening, required special education during early childhood, and was diagnosed with bilateral profound hearing loss by 8 years of age. Physical examination revealed two preauricular appendages anterior to the right ear, without additional systemic abnormalities. The proband II 1's parents had normal hearing and no family history of hearing loss, but declined genetic testing, precluding segregation analysis (*Figure 5A and B*).

The p.E182* variant introduces a premature stop codon at position Glu182, resulting in a truncated HSD17B7 protein (*Figure 5C*). The multiple sequence alignment showed that Glu182 is evolutionarily conserved across vertebrates, from zebrafish to humans (*Figure 5D*). According to UniProt annotation, HSD17B7 contains an extracellular domain (1–229 aa), a transmembrane region (230–250 aa), and a cytoplasmic domain (251–341 aa). The p.E182* variant is predicted to remove the transmembrane region and cytoplasmic domain (*Figure 5E*). Together, these analyses identify p.E182* as a candidate variant in HSD17B7 that may be associated with hearing loss.

## The HSD17B7$^{E182*}$ truncation failed to rescue MET activity and auditory-related behaviors in hsd17b7 mutants

To assess the functional impact of the p.E182* variant, human HSD17B7 or truncation (HSD17B7$^{E182*}$) mRNAs were microinjected into the fertilized one-cell stage embryos of *hsd17b7* mutants. Injection of HSD17B7 mRNA significantly increased the FM4-64 uptake in LLHCs of *hsd17b7* mutants, whereas the injection of HSD17B7$^{E182*}$ mRNA failed to restore FM4-64 labeling (*Figure 6A and B*), indicating that the truncated protein is unable to rescue normal MET activity. Consistently, acoustic startle response assays revealed that HSD17B7 mRNA rescued behavioral responses of hsd17b7 mutants, as assessed by movement trajectory, swimming distance, and peak velocity. In contrast, the p.E182* variant showed no rescuing effect on these behavioral deficits (*Figure 6C–E*).

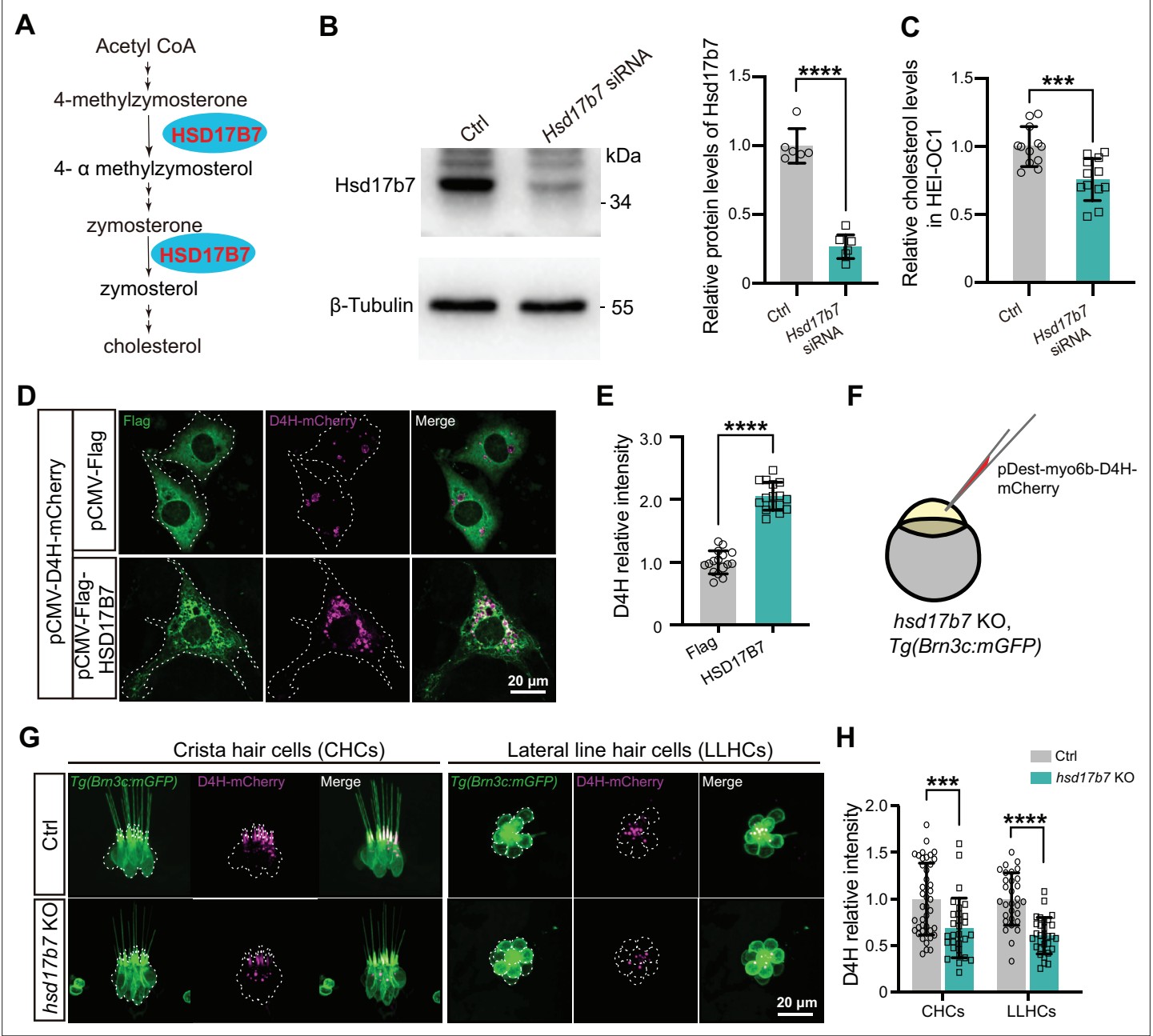

**Figure 4.** Hsd17b7 deficiency reduced cholesterol in hair cells. (**A**) Schematic illustration of the cholesterol biosynthesis pathway highlighting the role of HSD17B7. (**B**) Western blot analysis of Hsd17b7 protein levels in control and *Hsd17b7*-knockdown HEI-OC1 cells; quantification is shown on the right (n=6). Significance was determined by an unpaired two-tailed Student's *t*-test. ****p<0.0001, mean ± SD. (**C**) Quantification of the relative cellular cholesterol levels in control and Hsd17b7-knockdown HEI-OC1 cells (n=12). Significance was determined by an unpaired two-tailed Student's *t*-test. ***p<0.001, mean ± SD. (**D**) Representative immunofluorescence images showing the cholesterol probe D4H-mCherry in HEI-OC1 cells transfected with pCMV-Flag or pCMV-Flag–HSD17B7 plasmids. White dashed outlines indicate transfected cells. Scale bars, 20 µm. (**E**) Quantification of relative D4H-mCherry fluorescence intensity for (**D**) (n=16). Significance was determined by an unpaired two-tailed Student's *t*-test. ****p<0.0001, mean ± SD. (**F**) Experimental schematic illustrating expression of D4H-mCherry in control and *hsd17b7* knockout zebrafish larvae using the *Tg(Brn3c:mGFP)* background. (**G**) Representative confocal images of D4H-mCherry (magenta) in crista hair cells (CHCs) and lateral line hair cells (LLHCs) of control and *hsd17b7* knockout larvae at 4 dpf. White dashed outlines indicate hair cells. Scale bars, 20 µm. (**H**) Quantification of relative D4H-mCherry fluorescence intensity in vivo for (**G**) (n=30). Significance was determined by an unpaired two-tailed Student's *t*-test. ***p<0.001, ****p<0.0001, mean ± SD.

The online version of this article includes the following source data and figure supplement(s) for figure 4:

**Source data 1.** Original files for western blot analysis displayed in *Figure 4B*.

**Source data 2.** Original western blot analysis displayed in *Figure 4B*, labelled.

*Figure 4 continued on next page*

*Figure 4 continued*

**Figure supplement 1.** Knockdown of *hsd17b7* decreases cholesterol levels in hair cells.

## The HSD17B7$^{E182*}$ truncation disrupts intracellular cholesterol distribution and MET activity

To explore the potential contribution of HSD17B7$^{E182*}$ truncation to the patient's hearing loss, we compared the cellular activation of the wild-type and truncated HSD17B7 in vitro and in vivo. We analyzed the subcellular localization of Flag-tagged HSD17B7 and HSD17B7$^{E182*}$ in HEI-OC1 cells. Since cholesterol synthesis mainly occurs in the endoplasmic reticulum (ER; *Lev, 2012*; *Stevenson et al., 2016*), HSD17B7 was expected to localize to this compartment. Consistent with this expectation, immunostaining in HEI-OC1 cells showed that Flag-tagged HSD17B7 co-localized with the ER marker Calnexin. In contrast, HSD17B7$^{E182*}$ failed to localize to the ER and instead displayed a spotlike aggregate distribution (*Figure 7A*), a pattern further supported by fluorescence intensity profile analyses (*Figure 7B and C*).

Given that the patient is heterozygous for the variant, we next examined whether HSD17B7$^{E182*}$ interferes with HSD17B7 subcellular localization. Co-expression of Flag-HSD17B7 and HSD17B7$^{E182*}$-Myc in HEI-OC1 cells showed that HSD17B7 retained its ER localization, indicating that the truncated protein does not significantly alter the subcellular distribution of the wild-type protein (*Figure 7D and E*).

To further investigate the function of HSD17B7$^{E182*}$ expression in vivo, we overexpressed human HSD17B7 or HSD17B7$^{E182*}$ in zebrafish HCs. The FM4-64 uptake assay revealed that the overexpression

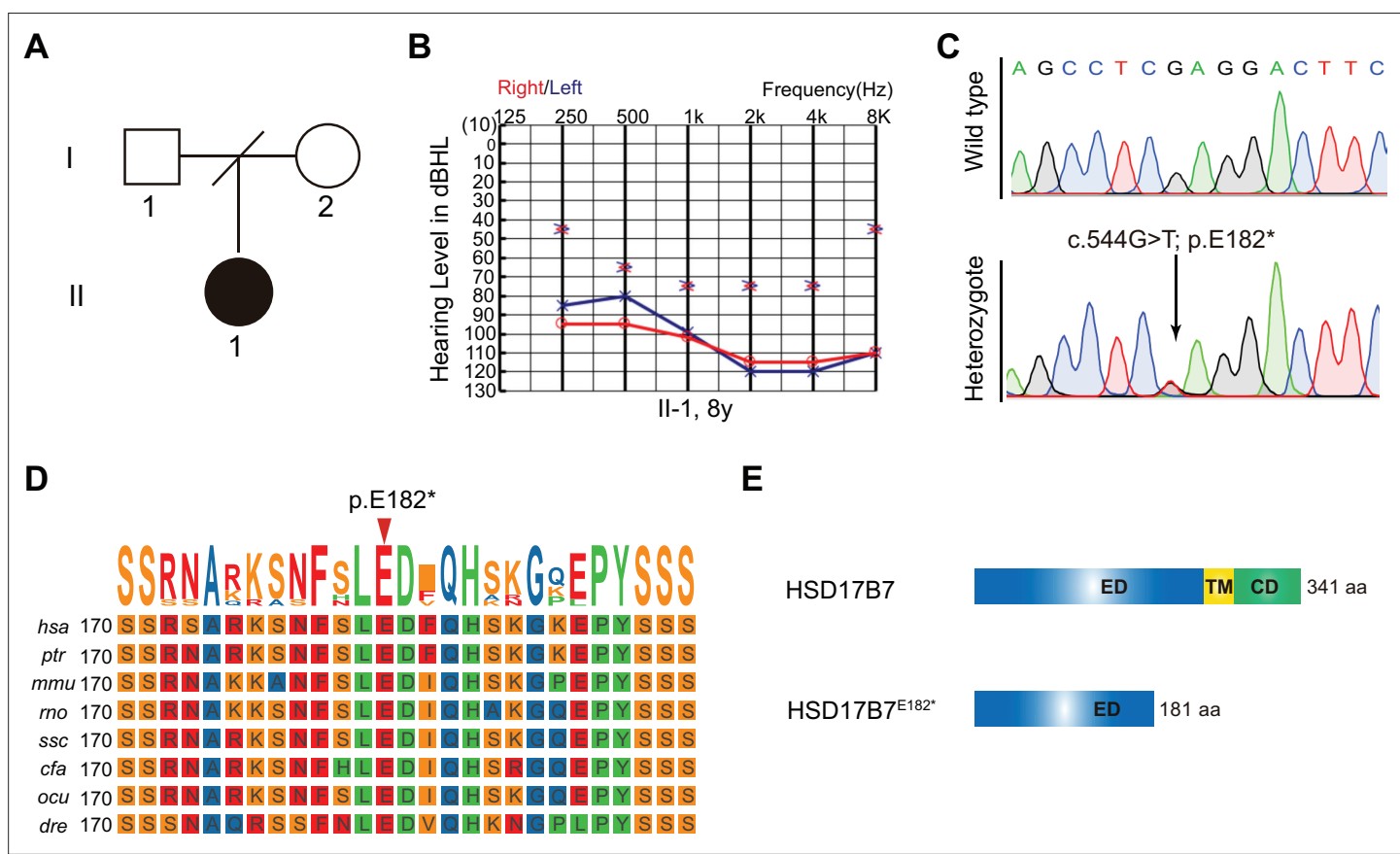

**Figure 5.** Identification of a heterozygous nonsense variant in human *HSD17B7*. (**A**) Two-generation family pedigree for the affected individual. The hearing-impaired individual is indicated by a black circle (female). (**B**) Pure-tone audiometry audiograms for the hearing-impaired individual at 8 years old. Blue represents the results for the left ear and red for the right ear. The affected individual shows severe-to-profound or profound HI. (**C**) Sequence of HSD17B7 mRNA in wild type and heterozygote. (**D**) Multiple sequence alignment of 8 different species using the TBtools program. The p.Glu182 residue, as indicated by a red arrow, is evolutionarily conserved from zebrafish to human. (**E**) The domain structure of human HSD17B7 and HSD17B7$^{E182*}$. ED, extracellular domain; TM, transmembrane; CD, cytoplasmic domain. The residue numbers are labeled at right.

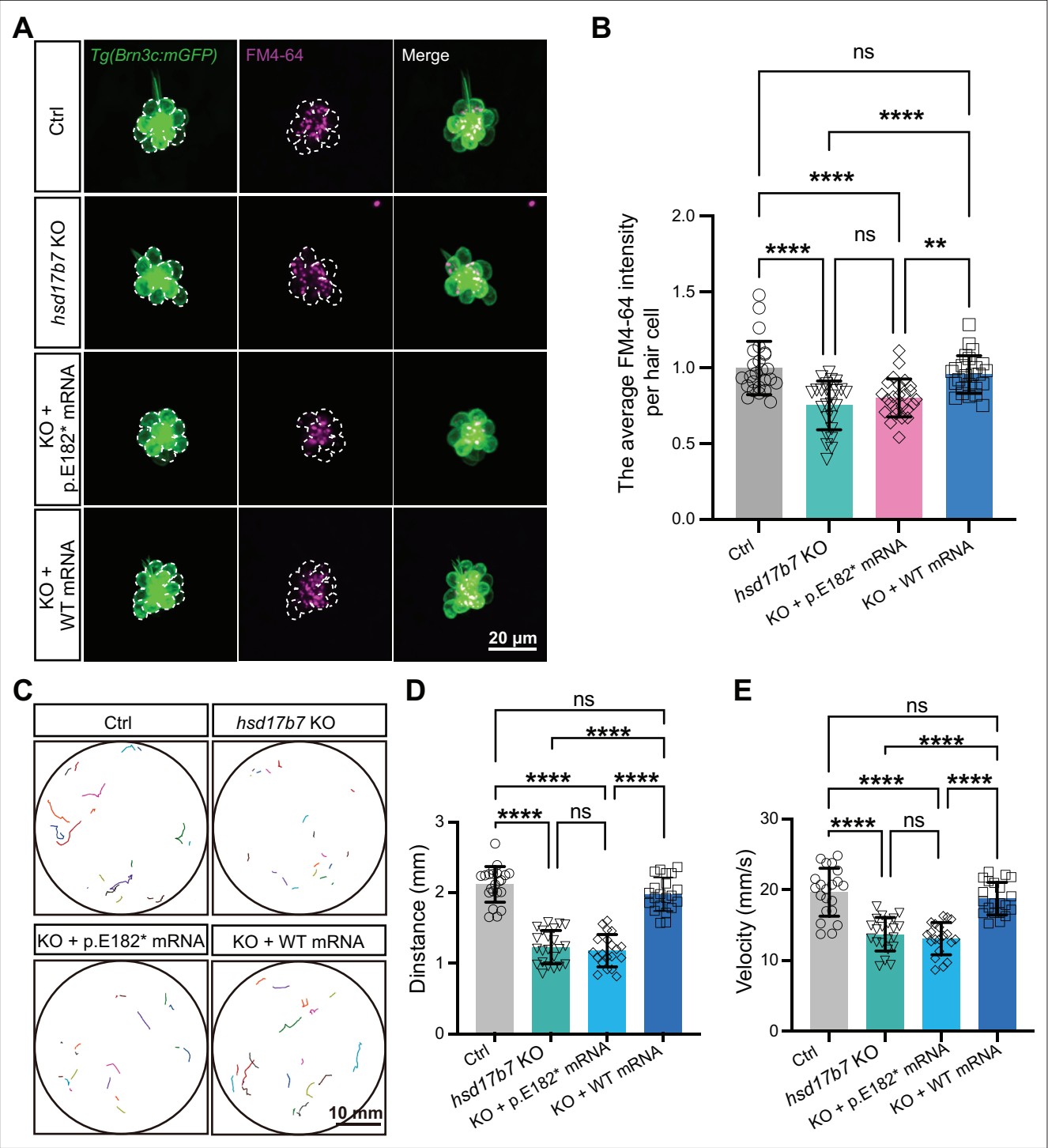

**Figure 6.** Wild-type but not HSD17B7^E182* restored MET activity and startle responses in hsd17b7 mutants. (**A**) Representative images of lateral line hair cells (LLHCs, green) and FM4-64–labeled functional HCs (magenta) in single neuromasts of *Tg(Brn3c:mGFP)* at 5 dpf from control, *hsd17b7* knockout (KO), KO injected with *HSD17B7^E182** (p.E182*) mRNA, and KO injected with *HSD17B7* (WT) mRNA groups. White dashed outlines indicate LLHCs. (**B**) Quantification of the relative FM4-64 intensity per HC (n=25). One-way ANOVA followed by Tukey's multiple comparisons, **p<0.01, ****p<0.0001, ns, non-significant (p>0.05), mean ± SD. (**C**) Representative locomotor trajectories of 5 dpf larvae exhibiting behavioral responses to a single acoustic stimulus (9 dB re. 1 m·s⁻², 60 Hz tone burst) in the indicated groups. (**D, E**) Quantification of total movement distance (**D**) and peak swimming velocity (**E**) in response to acoustic stimulation (n=20). One-way ANOVA followed by Tukey's multiple comparisons, ****p<0.0001, ns, non-significant (p>0.05), mean ± SEM.

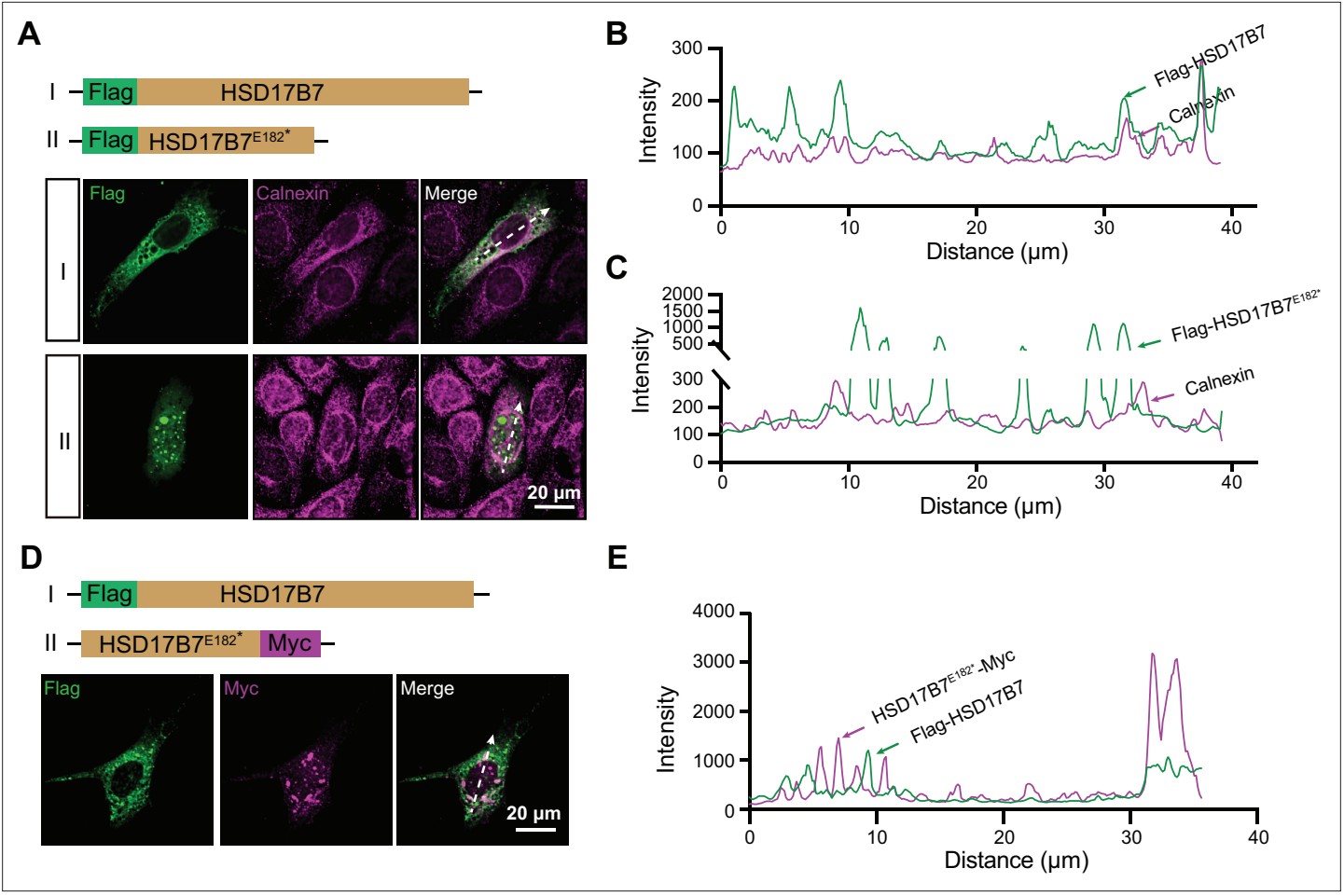

**Figure 7.** Subcellular localization of wild-type and p.E182* mutation HSD17B7. (**A**) Immunostaining shows the subcellular localization of Flag-HSD17B7 and Flag-HSD17B7$^{E182*}$ in HEI-OC1 cells. Calnexin was used as an endoplasmic reticulum (ER) marker. White arrows indicate the line-scan positions used for intensity profile analysis in (**B**) and (**C**). Scale bars, 20 μm. (**B, C**) Line-scan intensity profiles showing the spatial distribution of HSD17B7 (**B**) or HSD17B7$^{E182*}$ (**C**) relative to the ER marker Calnexin along the indicated axes in (**A**). (**D**) Representative immunofluorescence image of a HEI-OC1 cell co-expressing HSD17B7 and HSD17B7$^{E182*}$. White arrows indicate the positions used for line-scan analysis. Scale bars, 20 μm. (**E**) Line-scan intensity profiles showing the intracellular distribution of HSD17B7 and HSD17B7$^{E182*}$ along the indicated axis in (**D**).

of HSD17B7 did not significantly affect FM4-64 fluorescence intensity in LLHCs compared to the control. In contrast, the expression of HSD17B7$^{E182*}$ led to a significant reduction of fluorescence intensity (*Figure 8A and B*). Consistently, startle response assays showed that HSD17B7 overexpression showed comparable movement trajectories, distance, and velocity in response to auditory stimulation, whereas larvae expressing HSD17B7$^{E182*}$ reduced startle-associated behaviors (*Figure 8C–E*). These data suggest that expression of HSD17B7$^{E182*}$ negatively impacts auditory-related behavior in vivo.

Given our findings that Hsd17b7 regulates cholesterol homeostasis (*Figure 4*) and previous research that proper cholesterol distribution is critical for HC function (*Gao et al., 2022*), we next examined whether the HSD17B7$^{E182*}$ truncation disrupts intracellular cholesterol organization. HEI-OC1 cells were co-transfected with cholesterol probe D4H-mCherry together with pCMV-Flag, pCMV-Flag-HSD17B7, or pCMV-Flag-HSD17B7$^{E182*}$. While cholesterol distribution appeared as a diffuse pattern around the nucleus in the control and HSD17B7-expressing cells, HSD17B7$^{E182*}$ resulted in a spot-like aggregation of D4H-mCherry (*Figure 8F*). Line-scan intensity analysis further confirmed the aberrant colocalization of HSD17B7$^{E182*}$ with cholesterol-enriched compartments (*Figure 8G–I*).

To validate these observations in vivo, HSD17B7 or HSD17B7$^{E182*}$ was expressed in zebrafish HCs by injecting pDest-myo6b-HSD17B7-EGFP or pDest-myo6b-HSD17B7$^{E182*}$-EGFP plasmids into the *Tg(myo6b: D4H-mCherry)* transgenic line (*Figure 8J*). In HCs expressing HSD17B7, the protein

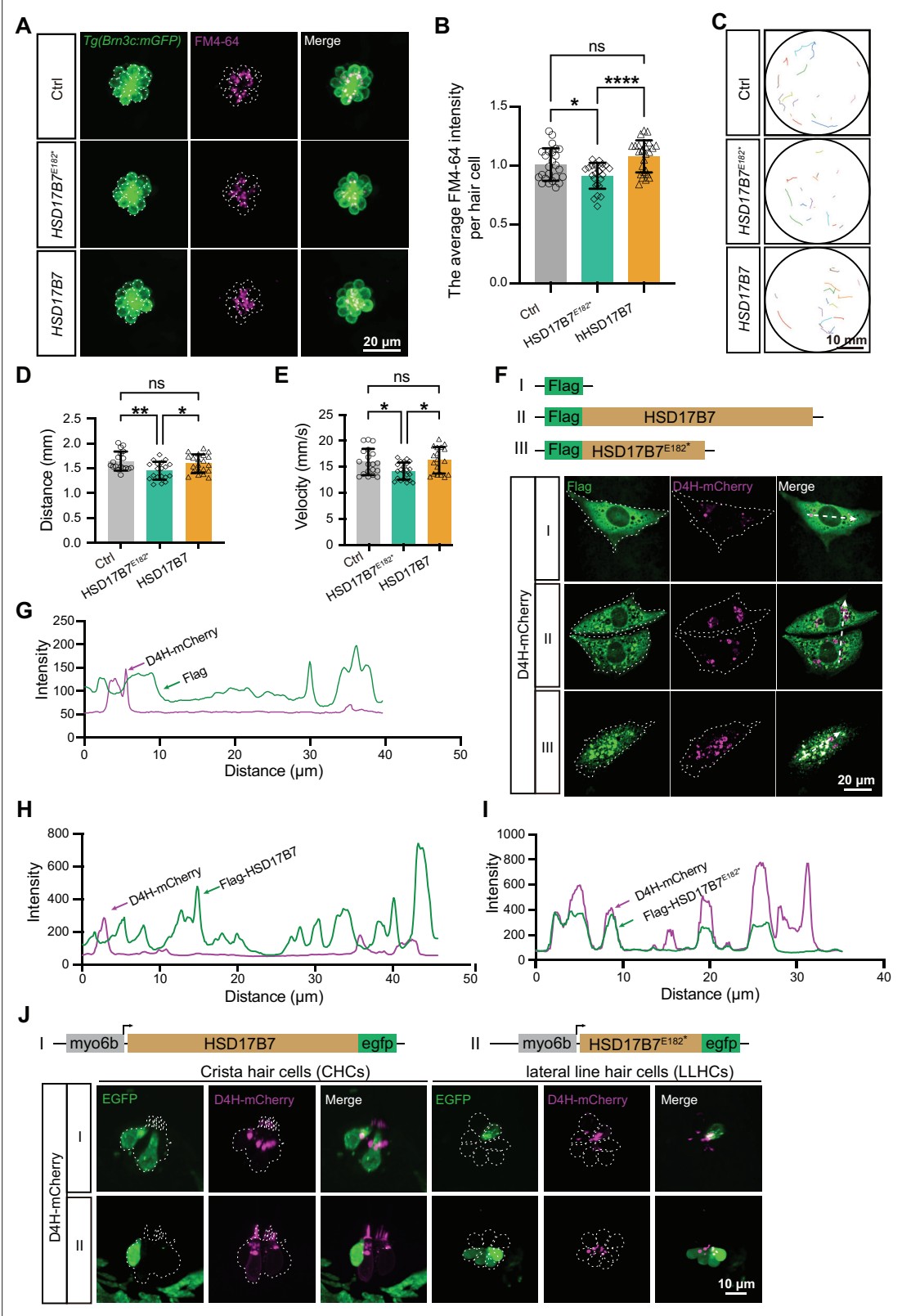

**Figure 8.** Expression of HSD17B7$^{E182*}$ impaired mechanotransduction activity, auditory-related behavior, and cholesterol organization. (**A**) Representative images of lateral line hair cells (LLHCs, green) and FM4-64–labeled functional hair cells (magenta) in single neuromasts of *Tg(Brn3c:mGFP)* larvae at 5 dpf following injection of HSD17B7 or HSD17B7$^{E182*}$ mRNA. The white dashed indicates LLHCs. Scale bars, 20 μm. (**B**) Quantification of the FM4-64 relative intensity per LLHCs (n=25). One-way ANOVA followed by Tukey's multiple comparisons, *p<0.05, ****p<0.0001, ns, non-significant (p>0.05),

*Figure 8 continued on next page*

*Figure 8 continued*

mean ± SD. (**C–E**) Moving traces and quantification of 20 embryos were filmed in 150 frames before and after stimulation (9 dB re. 1 m·s⁻², 60 Hz tone burst). Scale bars, 10 mm. One-way ANOVA followed by Tukey's multiple comparisons, *p<0.05, **p<0.01, ns, non-significant (p>0.05), mean ± SEM. (**F**) Immunostaining of the cholesterol probe D4H in HEI-OC1 cells transfected with pCMV-Flag, pCMV-Flag-HSD17B7, and pCMV-Flag-HSD17B7$^{E182*}$ constructs, respectively. White dashed outlines indicate transfected cells; arrows denote line-scan positions used for intensity profile analysis. Scale bars, 20 μm. (**G–I**) Line-scan intensity profiles showing the spatial distribution of Flag, HSD17B7, or HSD17B7$^{E182*}$ relative to D4H signal along the indicated axes in (**F**). (**J**) Representative images of crista and lateral line hair cells in *Tg(myo6b: D4H-mCherry)* larvae following expression of myo6b-driven HSD17B7-EGFP or HSD17B7$^{E182*}$-EGFP at 4 dpf. White dashed outlines indicate HCs. Scale bars, 10 μm.

exhibited a punctate intracellular distribution and did not noticeably alter the D4H-mCherry cholesterol signal compared with neighboring non-expressing HCs. In contrast, HSD17B7$^{E182*}$ showed a diffuse localization throughout the cytoplasm and nucleus, accompanied by a marked reduction or near-complete loss of the D4H-mCherry signal in the stereocilia position. These data suggest that HSD17B7$^{E182*}$ with aberrant subcellular localization may bind cholesterol and alter its intracellular distribution. Altogether, these results demonstrate that HSD17B7$^{E182*}$ has a negative effect by altering cholesterol distribution in HCs, thereby compromising MET function and impaired startle responses.

## HSD17B7$^{E182*}$ truncation disrupted the interaction with RER1

To further explore the pathological consequences of HSD17B7$^{E182*}$, we compared the interaction landscapes of wild type and mutant in HEI-OC1 cells (*Figure 9A*). Proteomic analysis of co-immunoprecipitated complexes revealed distinct interaction profiles between HSD17B7 and HSD17B7$^{E182*}$ (*Figure 9A–C*; *Figure 9—figure supplement 1*). Proteins uniquely associated with HSD17B7 were strongly enriched for cholesterol metabolic processes, whereas those preferentially associated with HSD17B7$^{E182*}$ were enriched for chromatin remodeling–related pathways.

Consistent with their subcellular localizations, cellular component analysis showed that HSD17B7-interacting proteins were predominantly enriched in the ER, whereas HSD17B7$^{E182*}$-associated proteins were enriched in the nucleus and cytoplasm (*Figure 9D*), in agreement with immunostaining results (*Figure 7A–C*). Among ER-associated interactors, RER1 emerged as the top HSD17B7-specific binding partner (*Figure 9E*). RER1 is known to mediate ER retention of membrane-associated proteins (*Yamasaki et al., 2014*; *Kaether et al., 2007*). Direct binding between HSD17B7 and RER1 was confirmed by in vitro binding assays, whereas the HSD17B7$^{E182*}$ failed to interact with RER1 (*Figure 9F*). Immunofluorescence analysis further demonstrated robust co-localization of RER1 with HSD17B7 but not with HSD17B7$^{E182*}$ (*Figure 9G–I*), indicating that the p.E182* mutation disrupts RER1-mediated ER retention. Given that HSD17B7 is normally localized to the ER, loss of RER1 interaction provides a mechanistic explanation for the aberrant subcellular distribution of the mutant protein.

In addition to altered localization, the p.E182* mutation markedly reduced HSD17B7 expression levels. Compared with HSD17B7, the mutant protein exhibited substantially decreased abundance, accompanied by a significant reduction in mRNA levels (*Figure 9—figure supplement 2A–C*). mRNA stability assays revealed a shortened half-life of HSD17B7$^{E182*}$ transcripts relative to wild type (*Figure 9—figure supplement 2D and E*), indicating that the nonsense mutation compromises transcript stability and consequently reduces protein expression.

## Discussion

This study reveals that HSD17B7 is enriched in sensory HCs in zebrafish and mice. Using the zebrafish model, we show that loss of *hsd17b7* markedly reduces cholesterol levels in HCs, leading to impaired MET function and abnormal hearing behaviors. These findings identify HSD17B7 as a previously unrecognized regulator of HC physiology and a candidate gene for sensory hearing loss. In addition to the animal models, we identified a previously undescribed heterozygous nonsense variant in HSD17B7 (c.544G>T, p.E182*) in a patient with sporadic deafness. Functional analyses revealed that the residual HSD17B7$^{E182*}$ protein alters subcellular localization and disrupts normal intracellular cholesterol distribution. Moreover, the mutation reduces mRNA half-life, decreases mRNA abundance, and significantly lowers protein expression. Collectively, these results suggest that the heterozygous c.544G>T (p.E182*) variant contributes to auditory dysfunction through potential pathogenic mechanisms: haploinsufficiency caused by reduced HSD17B7 expression and functional impairment due to altered

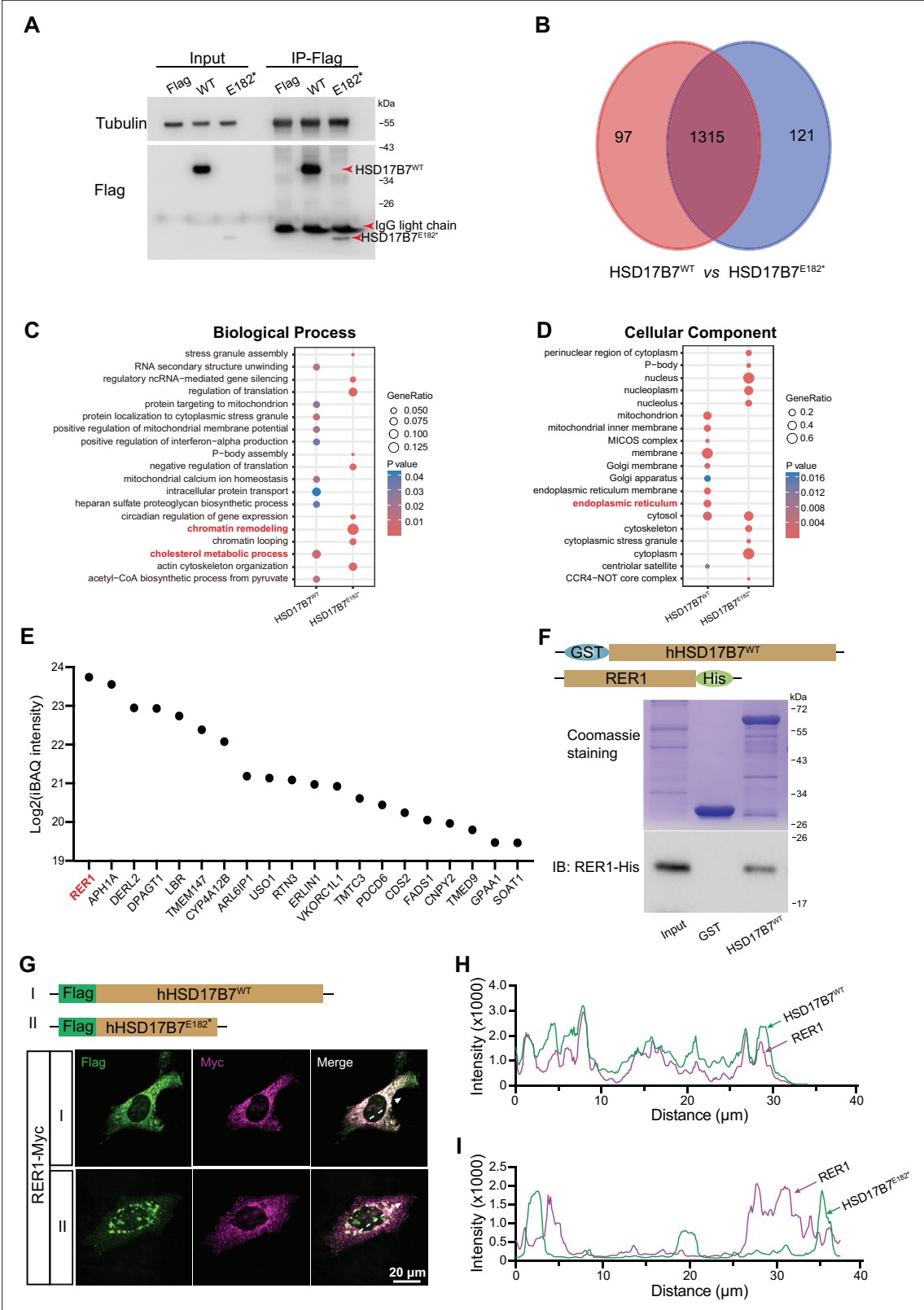

**Figure 9.** HSD17B7 interacts with RER1 to ensure ER localization. (**A**) Co-immunoprecipitation of Flag-tagged HSD17B7 and HSD17B7[E182*] from HEI-OC1 cells transfected with pCMV-Flag, pCMV-Flag-HSD17B7, or pCMV-Flag-HSD17B7[E182*]. Cell lysates were immunoprecipitated using anti-Flag beads. Tubulin and IgG light chains served as loading and immunoprecipitation controls, respectively. (**B**) Venn diagram showing the overlap and uniqueness of interacting proteins identified for HSD17B7 (red) and HSD17B7[E182*] (blue) by LC–MS/MS analysis. (**C**) Gene Ontology (GO) Biological Process

*Figure 9 continued on next page*

*Figure 9 continued*

enrichment analysis of proteins specifically associated with HSD17B7 or HSD17B7[E182*]. (**D**) GO Cellular Component enrichment analysis of HSD17B7- and HSD17B7[E182*]-specific interacting proteins. (**E**) Top 20 ER-localized proteins (ranked by iBAQ intensity) specifically associated with HSD17B7. (**F**) In vitro GST pull-down assays demonstrating direct interaction between HSD17B7 and RER1. The upper panel shows Coomassie blue staining of purified GST and GST-HSD17B7 proteins; the lower panel shows immunoblot detection of RER1-His. (**G**) Immunofluorescence staining of RER1-Myc in HEI-OC1 cells co-transfected with Flag-HSD17B7 or Flag-HSD17B7[E182*]. White arrows indicate positions used for fluorescence intensity profiling. Scale bars, 20 μm. (**H, I**) Fluorescence intensity profiles showing co-localization of RER1 with HSD17B7 (**H**) but not with HSD17B7[E182*] (**I**).

The online version of this article includes the following source data and figure supplement(s) for figure 9:

**Source data 1.** Original files for Co-immunoprecipitation analysis displayed in *Figure 9A*.

**Source data 2.** Original Co-immunoprecipitation analysis displayed in *Figure 9A*, labelled.

**Source data 3.** Original files for pull-down assays displayed in *Figure 9F*.

**Source data 4.** Original pull-down assays displayed in *Figure 9F*, labelled.

**Figure supplement 1.** Schematic of the experimental process for immunoprecipitation and LC-MS/MS to identify HSD17B7 and HSD17B7[E182*] interacted proteins.

**Figure supplement 2.** E182* mutation decreased HSD17B7 protein levels, mRNA levels, and the half-life periods.

**Figure supplement 2—source data 1.** Original files for western blot analysis displayed in *Figure 9—figure supplement 2B*.

**Figure supplement 2—source data 2.** Original western blot analysis displayed in *Figure 9—figure supplement 2B*, labelled.

cholesterol distribution. These mechanistic insights highlight the critical role of cholesterol homeostasis in MET and auditory-related function.

HSD17B7 was first identified as a prolactin receptor-associated protein in rats (*Duan et al., 1996*) and was proposed to be involved in estradiol biosynthesis (*Nokelainen et al., 1998*; *Törn et al., 2003*). Subsequent biochemical studies demonstrated that HSD17B7 participated in cholesterol synthesis by catalyzing the reduction of zymosterone to zymosterol (*Ohnesorg et al., 2006*; *Seth et al., 2006*; *Ohnesorg and Adamski, 2006*). Consistent with this role, both in vivo and in vitro studies have shown that loss of HSD17B7 reduces cholesterol levels, whereas its overexpression increases cholesterol abundance. Previous studies have shown that HSD17B7 is expressed in the liver, heart, brain, eye, and ear (*Shehu et al., 2008*; *Marijanovic et al., 2003*; *Krazeisen et al., 1999*), and scRNA-seq datasets have suggested its expression in mouse vestibular HCs (*Jan et al., 2021*). Although the role of the *HSD17B7* gene in ovarian and breast cancer has been extensively explored in the literature (*Plourde et al., 2009*; *Song et al., 2006*; *Wang et al., 2017*; *Wang et al., 2015*; *Sang et al., 2019*), the role of the *HSD17B7* gene in auditory function remains unclear. Our results fill this gap by demonstrating that HSD17B7 is enriched in HCs, as supported by whole-mount in situ hybridization, immunostaining, and scRNA-seq analyses from our study and others (*Qian et al., 2022*; *Burns et al., 2015*; *Xu et al., 2022*; *Wilkerson et al., 2021*). Given that loss of *hsd17b7* causes abnormal auditory-related behaviors, these data establish HSD17B7 as a conserved and essential regulator of HC function.

Cholesterol is a fundamental component of biological membranes, contributing to both their structural integrity and functional properties (*Maxfield and van Meer, 2010*; *Subczynski et al., 2017*). Previous studies have shown that either excessive or insufficient cholesterol levels, as well as abnormal cholesterol distribution, are detrimental to the auditory system (*Ding et al., 2020*; *Yao et al., 2019*; *Xing et al., 2015*; *Crumling et al., 2012*; *Gao et al., 2022*; *Takahashi et al., 2016*; *Thoenes et al., 2015*). However, previous studies have largely focused on systemic or membrane cholesterol alterations and have not addressed the contribution of HC-intrinsic cholesterol biosynthesis. Smith–Lemli–Opitz syndrome (SLOS), caused by mutations in DHCR7, exemplifies how impaired cholesterol synthesis can lead to multisystem defects, including sensorineural hearing loss, although auditory function in these patients has not been extensively investigated (*Zalewski et al., 2021*). Our study revealed that HSD17B7, a cholesterol biosynthetic enzyme, is enriched in sensory HCs and plays a crucial role in maintaining cholesterol levels required for auditory-related function. It should be noted that some localization and cholesterol distribution analyses were performed using C-terminal EGFP-tagged HSD17B7 constructs, which may influence protein behavior. However, the central conclusions of this study are supported by complementary loss-of-function models, rescue experiments using untagged mRNA, and independent cholesterol perturbation assays, reducing reliance on any single overexpression or fusion-based approach.

Here, we found that loss of Hsd17b7 or expression of the truncated HSD17B7[E182*] variant reduced cholesterol abundance and disrupted its intracellular distribution, resulting in defective MET function and auditory-related impairment. In contrast, overexpression of wild-type HSD17B7 elevated cholesterol levels but did not impair auditory-related behavior, suggesting that HC function requires a minimum critical cholesterol threshold rather than being sensitive to moderate cholesterol excess. Freeze-fracture studies have shown that the stereociliary membrane is densely enriched in cholesterol (*Forge et al., 1988*), supporting the notion that cholesterol is a key determinant of hair bundle membrane properties. Cholesterol is known to stiffen biological membranes (*Chakraborty et al., 2020*), a biophysical feature that may influence MET channel gating (*Powers et al., 2014*; *Powers et al., 2012*). Consistent with this idea, cryo-EM studies have revealed conserved protein–lipid interactions within the TMC1/TMC2-containing MET channel complex (*Jeong et al., 2022*; *Clark et al., 2024*). Together with our findings, these observations support a model in which HSD17B7 regulates MET function by maintaining appropriate cholesterol abundance within HCs. How cholesterol perturbations caused by HSD17B7 deficiency or mutation mechanistically alter MET channel activity remains an important question for future studies.

Dominant non-syndromic hearing loss is a common form of hereditary deafness (*Zhao et al., 2014*; *Hildebrand et al., 2011*; *Vahava et al., 1998*; *Guan et al., 2020*; *Zhao et al., 2022*; *Tian et al., 2018*; *Li et al., 2021*; *Mi et al., 2021*; *Kubisch et al., 1999*; *Mencía et al., 2008*; *Zhang et al., 2011*; *Morell et al., 2000*), and de novo mutations are frequently observed in patients with a negative family history. In the present study, both parents of the proband exhibited normal hearing and declined genetic testing, suggesting that the HSD17B7 (c.544G>T, p.E182*) variant may represent a de novo dominant mutation. This interpretation is consistent with previous reports showing that homozygous deletion of Hsd17b7 in mice is embryonic lethal, whereas heterozygous phenotypes have not been systematically characterized (*Jokela et al., 2010*). Notably, the mutant mRNA failed to rescue auditory defects in hsd17b7 mutants, and overexpression of HSD17B7[E182*] in wild-type animals impaired startle response. Although additional clinical cases will be required to confirm pathogenicity, our functional analyses provide experimental support for a negative effect of the HSD17B7[E182*] variant. Together with the marked reduction in mRNA and protein stability, these findings suggest that the HSD17B7[E182*] variant primarily acts through loss of function, with the possibility that aberrant intracellular localization and misdistribution of cholesterol further exacerbate HC dysfunction.

In summary, our study identifies HSD17B7 as a critical regulator of cholesterol synthesis in sensory HCs and as an essential factor in normal MET and sound-evoked sensory responses. Additionally, we identified a sporadic nonsense mutation in the *HSD17B7* gene in patients with deafness and provided a mechanistic explanation. These findings uncover a previously unrecognized link between cholesterol biosynthesis and HC MET function and provide a molecular framework for exploring cholesterol-targeted therapeutic strategies for hearing loss.

# Materials and methods

## Mouse and zebrafish husbandry

C57BL/6 J mice were purchased from the Laboratory Animal Center of Nantong University and maintained in a barrier facility under a 12 hr light/12 hr dark cycle at 25°C. The day of vaginal plug observation was embryonic day 0.5 (E0.5), and the day of birth was defined as postnatal day 0 (P0). Zebrafish (*Danio rerio*) were raised and maintained at 28.5 °C. Wild-type AB strain and the transgenic line *Tg(Brn3c: mGFP)* were used in the study described in our previous study (*Wei et al., 2022*). The transgenic line *Tg(myo6b: D4H-mCherry)* was generated in this work. The embryonic stage is defined as described in the literature (*Kimmel et al., 1995*). Embryos were collected following natural spawns and cultured in E3 medium (5 mM NaCl, 0.17 mM KCl, 0.33 mM CaCl$_2$, and 0.33 mM MgSO$_4$, pH 7.2). After 20 hpf, embryos were incubated in E3 medium containing 0.2 mM phenylthiourea (PTU) to prevent pigmentation for in situ hybridization and imaging. All animal experiments were performed under the guidelines of the Institutional Animal Care and Use Committee of Nantong University (No. S20240428-004).

## Human subjects and ethical approval

The study was approved by the Ethics Committee of West China Hospital of Sichuan University (No. 2020–606). Written informed consent was obtained from all participants or their legal guardians prior to inclusion in the study.

## Cell culture

The mouse auditory cell line HEI-OC1 was obtained from Prof. Renjie Chai and authenticated by short tandem repeat (STR) profiling (Shanghai Biowing Applied Biotechnology Co., Ltd). The cell lines were routinely tested negative for mycoplasma using the MycoBlue Mycoplasma Detection Kit (Vazyme Biotech, #D101-01), and cell aliquots from early passages were used. HEI-OC1 cells were cultured in DMEM (Wisent, #319–005 CL) medium supplemented with 10% FBS (Sigma-Aldrich, #F8318) in an incubator with 10% $CO_2$ at 33°C.

HEK293T (sex unknown) cells were obtained from the Shanghai Institute of Biochemistry and Cell Biology (SIBC) and authenticated by short tandem repeat (STR) profiling (Shanghai Biowing Applied Biotechnology Co., Ltd). The cell lines were routinely tested negative for mycoplasma using the Myco-Blue Mycoplasma Detection Kit (Vazyme Biotech, #D101-01), and cell aliquots from early passages were used. Cells were cultured in DMEM (Wisent, #319–005 CL) medium supplemented with 10% FBS (Sigma-Aldrich, #F8318) in an incubator with 5% $CO_2$ at 37°C.

## Antibodies

Primary antibodies used in this study included: anti-MYO7A monoclonal antibody (1:150, Developmental Studies Hybridoma Bank, #138–1); anti-HSD17B7 polyclonal antibody (1:150-1:1000, proteintech, #14854–1-AP); anti-Parvalbumin monoclonal antibody (1:500, sigma, #sab4200545); anti-MYC tag polyclonal antibody (1:200, proteintech, #16286–1-AP); anti-His antibody (1:2500, proteintech, #66005–1-Ig); anti-DYKDDDDK tag mouse monoclonal antibody (1:200-1:5000, proteintech, #66008–4-Ig); anti-calnexin polyclonal antibody (1:100, proteintech, #10427–2-AP); tubulin monoclonal antibody (1:5000, proteintech, #66031–1-Ig); goat anti-rabbit IgG(H+L) HRP (1:5000, MULTI SCIENCES, #GAR007); goat anti-mouse IgG(H+L) HRP (1:5000, MULTI SCIENCES, #GAM007); FITC goat anti-mouse IgG (H+L) (1:200, ABclonal, #AS001); Cy3 goat anti-rabbit IgG (H+L; 1:200, ABclonal, #AS007); anti-Digoxigenin-AP Fab fragments (1:1000, Roche, #11093274910).

## RNA isolation, reverse transcription (RT), and quantitative real-time PCR (RT-qPCR)

Total RNAs were extracted using TRIzol reagent (Invitrogen, #15596026) and treated with DNase I (Vazyme, #EN401) to remove genomic DNA contamination. RNA concentration was measured using a NanoDrop ND-2000 (Thermo Fisher Scientific, USA), and integrity was verified by agarose gel electrophoresis.

The cDNA was synthesized by using HiScript III RT SuperMix (Vazyme, #R323-01) following the manufacturer's directions. Subsequently, quantitative PCR was performed using the ChamQ SYBR qPCR Master Mix (Vazyme, #Q341-02) with the specified primers (*HSD17B7* primer, F: 5'- GACA AGCTTGGATCCATGCGAA –3', R: 5'- ACCTGGACAA TGGTGACCTC –3'; *Gapdh* primer, F: 5'-CACA GTCAAGGCCGAGAATGGGAAG-3', R: 5'- GTGGTTCACACCCATCACAAACATG) with a final volume of 20 µL under the following conditions: 15 min at 50 °C, 5 min at 95 °C, and then 40 cycles at 95 °C for 15 s and 60 °C for 30 s. Relative expression levels for the *hsd17b7* gene were calculated using the $2^{-\triangle\triangle CT}$ method, normalized to the *gapdh*. All reactions were repeated in triplicate for each sample. The results were analyzed using the GraphPad Prism software (version 9.4.0).

## Whole-mount in situ hybridization

The whole-mount in situ hybridization (WISH) of zebrafish was performed according to the following standard procedures. A 403 bp cDNA fragment of the zebrafish *hsd17b7* gene or a 472 bp cDNA fragment of the zebrafish *tmc1* gene was amplified via PCR using designed primers (*hsd17b7*-F: 5'-GACG TCCTCCAGTAATGCCC-3', *hsd17b7*-R: 5'-CATCTTGCTTGGT CGGGTGT-3'; *tmc1*-F: 5'-TTGGGCAG TGATGTGCTGTA-3', *tmc1*-R: 5'-GATGCTGTTT CTGCGTTGCT-3') and was cloned into the pGEM-T-easy vector. After linearization of the pGEM-T-easy vector, inserting the *hsd17b7* or *tmc1* fragment, the DIG RNA Labeling Kit (SP6; Roche, #11175025910) was used to prepare digoxigenin-labeled

*hsd17b7* or *tmc1* antisense mRNA probes through transcription in vitro. Subsequently, embryos at different developmental stages were hybridized with an *hsd17b7* or *tmc1* mRNA probe overnight after a series of treatments, including fixation in 4% paraformaldehyde in PBS, digestion in proteinase K (Roche, #3115879001), and incubation with a pre-hybridized mix. Finally, the alkaline phosphatase (AP)-conjugated antibody against digoxigenin (Roche, #11093274910) and the AP-substrate NBT/BCIP solution (Roche, #11681451001) were used to detect the *hsd17b7* or *tmc1* expression. The WISH images of zebrafish were acquired using a stereomicroscope (Olympus, MVX10, Japan).

## Plasmids construction

The HSD17B7 and RER1 full-length cDNAs were PCR amplified from the HEK293T cDNA and cloned into the pGEX-TEV and pET-23b vectors, respectively. HSD17B7$^{E182*}$ was generated by site-directed mutagenesis. For cell transfection, the cDNA of HSD17B7 and HSD17B7$^{E182*}$ were subcloned into the pCMV-Flag vector, respectively. The cDNA of RER1 and HSD17B7$^{E182*}$ were subcloned into the pcDNA3.1-Myc-His A vector, respectively. eGFP was subcloned into pCMV-Flag-HSD17B7 and pCMV-Flag-HSD17B7$^{E182*}$, respectively. pCS2-D4H-mCherry was obtained from G. Peng's lab. For plasmid injection of zebrafish, the *myo6b* promoter was subcloned into the pDestTol2 vector, followed by D4H-mCherry subcloned into the pDestTol2 vector. The cDNAs encoding the human protein of HSD17B7-eGFP and HSD17B7$^{E182*}$-eGFP were inserted into the p-mTol2-myo6b vector, then linking P2A-D4H-mCherry by overlap extension. For mRNA injection of zebrafish, the cDNAs encoding the human protein of HSD17B7 and HSD17B7$^{E182*}$ were inserted into the pCS2$^+$, and likewise, the cDNAs encoding the zebrafish full-length protein of Hsd17b7 were inserted into the pCS2$^+$, which were linearized using NotI and then transcribed in vitro to generate mRNA. All constructs were verified by DNA sequencing.

## Cell transfection

HEI-OC1 cells were transfected with plasmids using an X-treme GENE HP DNA transfection reagent (Roche, #6366236001) according to the manufacturer's protocol. After 36 hr or 48 hr of transfection, the cells were fixed or collected and subjected to further analyses, including western blots, Co-IP, IF, or RT-qPCR.

HEI-OC1 cells were transfected with small interfering RNA. The mouse *Hsd17b7* siRNA oligos (F: 5'-GGAGGUGUUUGAAACCAAUTT-3', R: 5'-AUUGGUUUCAAACACCUC CTT-3') were synthesized from GenePharma (Shanghai, China). Cells were transfected with siRNA oligos to knock down mHSD7B7 using Lipofectamine RNAiMAX (Invitrogen, #13778–150) according to the manufacturer's protocols. Approximately $1\times10^5$ cells per well were placed in six-well plates. When the cell density reached 70–80%, 4 μL of control siRNA or mHsd17b7 siRNA was added to 150 μL of opti-MEM (Gibco, #31985070), then 5 μL Lipofectamine RNAiMAX was added, incubated at room temperature for 5 min, and then added to the cell culture medium. After 48 hr of transfection, the transfected cells were harvested for western blots or cholesterol assay.

## mRNA stability assay

HEI-OC1 cells were transfected with plasmids using an X-treme GENE HP DNA transfection reagent. After 24 hr, the cells were treated with 10 μM actinomycin D (MCE, #HY-17559) for 0, 2, 4, 6, and 8 hr. Then, the collected cells and isolated RNA were used for RT-qPCR (*Ratnadiwakara and Änkö, 2018*).

## Immunostaining and image acquisition

Cultured cells grown on coverslips, after 36 hr of transfection, were fixed with 4% paraformaldehyde in PBS for 30 min at room temperature and washed three times with PBS, then permeabilized in 0.3% PBST (0.3% Triton X-100 in PBS) for 20 min. After three times washes with PBS, the cells were blocked with 10% FBS in PBS, followed by incubation with anti-primary antibodies overnight at 4 °C, and then incubated with fluorescent dye-labeled secondary antibodies for 2 hr at room temperature before mounting using the mounting solution with DAPI (1:500, SouthernBiotech, #011–20).

The dissected mouse cochleae were fixed in 4% paraformaldehyde in PBS for 1 hr at room temperature. After three times washes with 0.1% PBST, the cochleae were briefly blocked with a blocking medium (PBS containing 1% Triton X-100 and 10% heat-inactivated donkey serum, pH 7.2) for 1 hr at room temperature, followed by incubation with the anti-primary antibody overnight at 4 °C. The

samples were then washed in 0.1% PBST and incubated with fluorescent dye-labeled secondary antibodies for 2 hr at room temperature, before mounting in mounting solution containing DAPI (1:500, SouthernBiotech, #011–20).

The immunostaining images of stained HEI-OC1 cells and cochlea were acquired by a Nikon confocal microscope with NIS-Elements software. For zebrafish live imaging, the larvae were anesthetized with tricaine MS-222 (Sigma, #A5040) and mounted in 0.6% low-melting agarose (Invitrogen, #16520050) with a lateral view. All reconstructed three-dimensional images and contrast adjustments were processed using Imaris X64 (version 9.0.1).

## FM4-64 labeling
To investigate the basal activity of HCs, the FM4-64 (1:500, Invitrogen, #T13320) vital dye was used to specifically label functional HCs in the neuromasts. The staining procedures were carried out as described previously (*Maeda et al., 2017*; *Kindt et al., 2012*; *Meyers et al., 2003*; *Lai et al., 2022*). The free-swimming larvae were incubated in 3 µM FM4-64 vital dye for 15 s at room temperature in the dark. Afterward, the fish were rinsed three times using a PTU medium and imaged. The images of stained HCs were acquired by a Nikon confocal microscope with NIS-Elements software.

## Measurement of total cholesterol
HEI-OC1 cells were homogenized in chloroform/methanol (2:1) to extract lipids and then centrifuged at $20,000 \times g$ for 10 min. The organic phase was harvested and dried using nitrogen flow. Total cellular cholesterol was quantified using an Amplex Red cholesterol assay kit (Invitrogen, #A12216) through a multilabel reader (PerkinElmer, USA) according to the manufacturer's instructions.

## Quantification of the relative fluorescence intensity of FM4-64 and D4H
Quantitative analysis of FM4-64 uptake in zebrafish LLHCs and D4H-mCherry fluorescence in HEI-OC1 cells was performed at the single-cell level. For zebrafish experiments, HCs were identified based on *Tg(Brn3c:mGFP)*. Individual HCs within each neuromast were manually segmented using ImageJ, and the mean FM4-64 fluorescence intensity per HC was measured. Background fluorescence was determined from adjacent cell-free regions and subtracted from each measurement. For each larva, the average FM4-64 intensity was calculated by averaging values from all analyzed HCs.

For D4H-mCherry cholesterol sensor analysis in HEI-OC1 cells, individual cells were segmented based on cell morphology, and the mean intracellular D4H-mCherry fluorescence intensity per cell was quantified. Background fluorescence was measured from cell-free regions within the same field of view and subtracted to normalize for imaging variability. Fluorescence intensities were then averaged across cells for each condition and used for statistical analysis. All fluorescence measurements were performed on raw images acquired under identical imaging settings across experimental groups.

## Morpholino and mRNA injections
For inhibiting the expression of *hsd17b7*, *hsd17b7*-specific splicing-blocking morpholinos were designed and procured from Gene Tools, Inc, and the precise sequence was (5'-TGCAAACAGGTA ACAAAACTGTGTG-3'). The morpholino powder was dissolved in RNase-free water to prepare the working solution at a final concentration of 0.3 mM for subsequent operations. About 2 nL dose of morpholino work solution was microinjected into zebrafish embryos at the one-cell stage. To assess morpholino efficiency, embryos injected with morpholino were collected, and RNA was extracted and reverse-transcribed into cDNA. The designed primers flanking exon 1 and exon 2 (F: 5'-TACA CAGGCAAACGTTAGAAGC-3', R: 5'-CTTGAACTCCTCTGCACCCTT-3') were used to amplify the fragment containing the mis-splicing target site, which was located at the connection of exon 1 and intron 1. For rescue experiments, exogenous mRNA was first transcribed in vitro. Briefly, the designed primers (zebrafish *hsd17b7* mRNA primer, F: 5'-CGCGGATCCATGAAGAAAGTAGTTTTGG T-3', R: 5'- CCGGAATT CTCACATTCCATTTCTTTCTT-3'; human *HSD17B7* mRNA primer, F: 5'-CGGG ATCCATGCGAAAGGTGGTTTTGATC-3', R: 5'-GCTCTAGATTATAGGCATGA GCCACTGA-3'; human *HSD17B7*$^{E182*}$ mRNA primer, F: 5'-CGGGATCCATGCGAAAG GTGGTTTTGATC-3', R: 5'-GCTCTAGA CTAGAGGCTGAAATTAGATT-3') were used to amplify target DNA containing the coding sequence. Subsequently, the pCS2 vector inserted into the amplified fragment was linearized as a template for mRNA transcription using the SP6 mMESSAGE mMACHINE Kit (Invitrogen, #AM1340). After purifying

using the RNeasy Mini Kit (QIAGEN, #74104), mRNA with a 70 ng/µL concentration was co-injected into one-cell stage embryos with morpholino for rescue experiments.

## sgRNA/Cas9 mRNA synthesis and injection

sgRNA was designed against the *hsd17b7* gene (ENSDARG00000088140) using the CRISPR design tool (https://www.crisprscan.org). The sgRNA specifically targeting exon 3 of *hsd17b7* (5'-GGGAATCA TGCCTAATCCCA-3') was first synthesized as follows. The GenCrispr sgRNA synthesis kit is used to generate a gRNA DNA template with a T7 promoter and to synthesize gRNA via in vitro transcription (Genscript Biotech Co., Ltd., Nanjing, China). The pXT7-zCas9 plasmid was linearized using XbaI and then transcribed in vitro to generate Cas9 mRNA using the mMESSAGE mMACHINE T7 kit (Invitrogen, #AM1344). At the one-cell stage, embryos were injected with a 2–3 µL solution containing 300 ng/µL Cas9 mRNA and 150 ng/µL sgRNA. The injected embryos were then grown in an E3 medium. At 24 hpf, genomic DNA was extracted from injected embryos, and potential CRISPR-induced mutations were identified by PCR. Primers used for identification were designed around the *hsd17b7* sgRNA target sites (F: 5'-AAAAACTTATTTTATTCCAGCCCAA-3', R: 5'-TTTCCAT GCAGCACTATCAAACA ATT-3').

## Startle response test

The acoustic startle reflex was performed as described previously (*Yang et al., 2017*; *Gong et al., 2021*). Briefly, a plastic plate attached to a mini vibrator was used to place 20 normal larvae at 5 days post-fertilization (dpf), while an infrared digital video tracking system was used to monitor their swimming behavior. 60 Hz tone bursts at two sound levels of 9 dB re. $1 \text{ m·s}^{-2}$ were applied to the amplifier to drive the vibrator. Acoustic vibration stimuli lasting 30 ms, with an inter-stimulus interval of 180 s, were set and applied. Each sound vibration stimulus level was repeated 20 times, and the locomotion behavior of the larvae with C-shape motion in response to this stimulus was recorded. Finally, the movement's typical parameters of mean distance and peak velocity were analyzed to assess the startle response of larvae to sound vibration stimuli.

## In vitro binding assay

GST and GST-HSD17B7 fusion proteins were immobilized on GSH resins and incubated with purified His-RER1 proteins overnight at 4°C and extensively washed with washing buffer (0.3% PBST). Bound proteins were separated using SDS-PAGE and visualized with Coomassie Blue staining and western blots.

## Western blots

Frozen cells were lysed in RIPA buffer (50 mM Tris-HCl, 150 mM NaCl, 1 mM EDTA, 1% sodium deoxy-cholate, 1% w/v protein inhibitor [Roche, #4693132001]) for 30 min and centrifuged at $12,000 \times g$ for 15 min at 4°C. Protein concentrations were determined using a BCA protein assay kit (Thermo Fisher, #23227).

The proteins were separated on SDS-PAGE gels and subsequently transferred to a PVDF membrane (Millipore, USA). The membranes were blocked in blocking buffer (1×TBS containing 0.5% milk and 0.5% Tween 20) for about 45 min at room temperature, then incubated overnight with primary anti-bodies in TBS containing 4% BSA, 1% Tween 20, and 0.05% $NaN_3$. The next day, the PVDF membranes were incubated with the appropriate HRP-conjugated secondary goat antibodies for 2 hr at room temperature after extensive washing with a blocking buffer. The blots were detected using ECL Western Blotting Detection Reagents (GE, #RPN2106), and the acquired images were analyzed using ImageJ (version 1.8.0).

## Co-IP, mass spectrum, and GO analysis

Cell lysates from pCMV-Flag, pCMV-Flag-HSD17B7, and pCMV-Flag-HSD17B7[E182*] transfected cells were incubated with 30 µL of beads conjugated to anti-Flag antibodies (Smart-Lifesciences, #SA042001), respectively, overnight at 4°C. The next day, Flag beads were washed three times with lysis buffer, and the immune complexes were eluted with 2× SDS sample buffer and subjected to three samples for SDS-PAGE and mass spectrum analysis.

For mass spectrum, the immunoprecipitated HSD17B7 and HSD17B7[E182*]-associated proteins were washed and dissolved in ammonium bicarbonate buffer (25 mM, pH 8.0), followed by trypsin digestion at 37°C for 16 hr. The nano-LC-MS/MS experiments were performed with LTQ-Orbitrap MS (Thermo Fisher, USA) equipped with a Nano electrospray ion source.

For the Venn diagram and GO analysis, microarray analysis identified HSD17B7 and HSD17B7[E182*]-specific associated proteins by mass spectrum (peptide ≥1, unique peptide ≥1, p-value <0.05). Cellular component GO term enrichment analysis was performed using the web-based DAVID software (*Huang et al., 2009*). Ontology networks were further investigated and visualized using R (version 4.4.1).

## Public scRNA-seq datasets and expression profiling of hsd17b7

Zebrafish scRNA-seq data were downloaded from NCBI Gene Expression Omnibus (GSE221471; *Qian et al., 2022*). The basic procedure for single-cell integrated data analysis was performed using Seurat 4.0.1 (*Stuart et al., 2019*). Cell clusters corresponding to LLHCs, MHCs, CHCs, supporting cells, and mantle cells were selected based on previously published annotations. Feature plots, violin plots, and average expression analyses were used to examine the expression pattern of *hsd17b7* across cell types. To assess dynamic changes in *hsd17b7* expression during HC differentiation, trajectory analysis was performed using Monocle 3 (v1.0.1). Cells were ordered along pseudotime, with mantle cells as the root population, and gene-expression changes were visualized along the inferred developmental trajectory.

Mouse scRNA-seq data were downloaded from previous articles (GSE71982, GSE168901, GSE202920; *Burns et al., 2015*; *Xu et al., 2022*; *Wilkerson et al., 2021*). The basic procedure for single-cell integrated data analysis and batch-effect correction was performed using Seurat 4.0.1 (*Stuart et al., 2019*). We selected four cell populations for re-analysis, including IHCs, OHCs, utricle hair cells, and CHCs.

## Single-cell RNA sequencing of control and hsd17b7 mutant hair cells

To investigate genotype-dependent transcriptional remodeling in sensory HCs, we performed single-cell RNA sequencing on fluorescence-activated cell sorting (FACS)-isolated HCs from control and *hsd17b7* mutant zebrafish.

## Isolation of hair cells and single-cell RNA sequencing

mGFP-positive HCs were isolated from control and *hsd17b7* mutant zebrafish larvae by FACS. Dissociated cells were gated to exclude debris and doublets, and viable mGFP[+] cells were collected for downstream analysis. Single-cell RNA sequencing libraries were generated using the 10×Genomics Chromium platform according to the manufacturer's instructions. Libraries were sequenced on an Illumina platform to obtain paired-end reads. Raw sequencing data were processed using the Cell Ranger pipeline (10×Genomics) to generate gene expression matrices for each sample. Downstream analyses were performed in R using the Seurat package.

## Data preprocessing, integration, and cell type annotation

Cells with low gene counts or high mitochondrial transcript proportions were excluded to remove low-quality cells. Gene expression matrices from control and *hsd17b7* mutant samples were normalized and integrated using Harmony to correct for batch effects while preserving biological variation.

Dimensionality reduction was performed using principal component analysis (PCA), followed by Uniform Manifold Approximation and Projection (UMAP) for visualization. Clustering was conducted using a shared nearest neighbor (SNN) graph-based approach. Cell clusters were annotated based on established marker genes for HCs and supporting cells. HCs were identified by robust expression of *myo6b*, while inner ear and LLHC subtypes were distinguished by *lhfpl5a* and *lhfpl5b*, respectively. Supporting cell populations were annotated using *stm* as a pan–supporting cell marker, with *otogl* and *irg1l* marking IE and LL supporting cells, respectively. Cell type annotations were validated using both violin plots and Nebulosa density visualizations.

## Pseudobulk differential expression analysis

To assess genotype-dependent transcriptional changes while accounting for cell-to-cell variability, pseudobulk differential expression analysis was performed for HCs. Gene expression counts from

individual cells were aggregated by genotype to generate pseudobulk profiles. Differential expression between control and *hsd17b7* mutant HCs was assessed using a generalized linear model framework. Genes with an adjusted p-value <0.05 were considered differentially expressed. Results were visualized using volcano plots, with upregulated and downregulated genes highlighted accordingly.

### Gene Ontology and pathway enrichment analysis

Differentially expressed genes were subjected to GO biological process enrichment analysis using curated annotation databases (*Yu et al., 2012*). Enrichment analyses were performed separately for upregulated and downregulated gene sets. Significantly enriched GO terms were visualized, and representative genes contributing to each term were illustrated using Sankey diagrams to highlight functional relationships between gene sets and biological processes.

GSEA was additionally performed to assess coordinated changes in cholesterol biosynthesis–associated pathways at the transcriptome-wide level. Ranked gene lists derived from pseudobulk differential expression results were used as input, and enrichment significance was assessed using normalized enrichment scores and adjusted p-values.

### Module score analysis of MET- and cholesterol-related gene sets

Module scores were computed using the AddModuleScore function in Seurat, based on normalized expression values from the RNA assay. Module score distributions were examined separately in LL_HCs and inner ear hair cells (IE_HCs). Differences between control and *hsd17b7* mutant samples were assessed using the Wilcoxon rank-sum test. Results were visualized using heatmaps, violin plots, and ridge plots. Ridge plots were generated using z-scored module scores to illustrate population-wide shifts in transcriptional states rather than the emergence of discrete subpopulations.

### Visualization and statistical analysis

UMAP feature plots, heatmaps, violin plots, and ridge plots were generated using Seurat and associated R visualization packages. Nebulosa density plots were used to visualize spatial expression patterns of marker genes across low-dimensional embeddings. Statistical analyses were performed using R, and exact p values or significance levels are reported in the corresponding figure panels and legends.

### Whole-exome sequencing and Sanger sequencing analyses

Whole-exome sequencing was performed in the proband II 1. Genomic DNA samples were extracted from whole blood samples of the proband II 1 (*Figure 5A*).

The exomes and flanking intronic regions from whole blood DNA samples were captured by Agilent SureSelect Human All Exon Kit (Agilent Technologies, USA). The captured DNA was sequenced on the Illumina HiSeq 4000 sequencing platform (Illumina, USA). Bioinformatics were aligned to the NCBI build37/hg19 assembly using the BWA (version 0.7.12) software. Each sample was covered to an average sequencing depth of at least 100×. SNPs and indels were identified using the GATK HaplotypeCaller software. Candidate pathogenic variants were defined as nonsense, missense, splice-site, and indel variants with allele frequencies of 0.001 or less in public variant databases (dbSNP, 1000 Genomes, ESP6500, nci60, GnomAD) and in disease databases (COSMIC, ClinVar, OMIM, GWAS). Genotypes distributed every 0.3 cM of genomic region were chosen for calculation of the logarithm of odds (LOD) scores using the Merlin v. 1.1.2 parametric linkage analysis package. The proband II 1 was genotyped by Sanger sequencing analyses with designed primers (F: 5'-GTACTCTGATTGGTGACGGGTGAG-3', R: 5'- GACAGTCATAGTTCATAGTTTATT-3').

### Quantification and statistical analysis

GraphPad Prism 9 (version 9.4.0) supported the whole statistical analysis. All data presented as mean ± SD or mean ± SEM. An unpaired two-tailed Student's t-test was performed for two-group comparisons, while multiple comparisons were illustrated using one-way ANOVA. The p-value less than 0.05 (p≤0.05) was considered significantly different. p≤0.05, p≤0.01, p≤0.001, and p≤0.0001 were symbolized with '*, **, ***, and ****', respectively, and 'ns' represented no significance, p>0.05.

## Acknowledgements

This work was supported by the National Natural Science Foundation of China Grants (32200783 to YQS; 92368104 and 32350017 to DL) and the Natural Science Foundation of Jiangsu Province Grants (BK20220607 to YQS).

## Additional information

### Funding

| Funder | Grant reference number | Author |
| --- | --- | --- |
| NSFC | 32200783 | Yuqian Shen |
| NSFC | 92368104 | Dong Liu |
| NSFC | 32350017 | Dong Liu |
| NSFJPG | BK20220607 | Yuqian Shen |
| NSFC | 82572137 | Yuqian Shen |

The funders had no role in study design, data collection and interpretation, or the decision to submit the work for publication.

### Author contributions

Yuqian Shen, Conceptualization, Funding acquisition, Investigation, Methodology, Writing – original draft, Writing – review and editing; Ziyang Wang, Conceptualization, Investigation, Methodology, Writing – original draft, Writing – review and editing; Xun Wang, Fuping Qian, Methodology; Mingjun Zhong, Resources, Methodology; Xin Wang, Formal analysis, Writing – original draft, Writing – review and editing; Jing Cheng, Resources, Supervision; Dong Liu, Conceptualization, Supervision, Funding acquisition, Investigation, Writing – original draft, Writing – review and editing

### Author ORCIDs

Yuqian Shen https://orcid.org/0000-0001-8296-525X
Ziyang Wang https://orcid.org/0009-0003-9470-8914
Xun Wang https://orcid.org/0009-0008-0289-6863
Fuping Qian https://orcid.org/0000-0001-5745-9304
Xin Wang https://orcid.org/0000-0002-5536-793X
Jing Cheng https://orcid.org/0009-0009-8746-0125
Dong Liu https://orcid.org/0000-0002-2764-6544

### Ethics

The study was approved by the Ethics Committee of West China Hospital of Sichuan University (No. 2020-606). Written informed consent was obtained from all participants or their legal guardians prior to inclusion in the study.

All animal experiments were performed under the guidelines of the Institutional Animal Care and Use Committee of Nantong University (approval no: S20240428-004).

Reviewer #1 (Public review): https://doi.org/10.7554/eLife.108108.3.sa1
Reviewer #2 (Public review): https://doi.org/10.7554/eLife.108108.3.sa2
Author response https://doi.org/10.7554/eLife.108108.3.sa3

## Additional files

### Supplementary files

MDAR checklist

## Data availability

The plasmids and zebrafish lines generated in this study are available on request to the Lead Contact. The published article includes all datasets generated or analyzed during this study. Any additional information required to reanalyze the data reported in this paper is available from the Lead Contact upon request. Sequencing data have been deposited in GEO under accession codes GSE319132.

The following dataset was generated:

| Author(s) | Year | Dataset title | Dataset URL | Database and Identifier |
| --- | --- | --- | --- | --- |
| Shen Y, Wang Z, Wang X, Qian F, Wang X, Liu D | 2026 | HSD17B7 is required for Sensory Hair Cells Function by Regulating Cholesterol Synthesis | https://www.ncbi.nlm.nih.gov/geo/query/acc.cgi?acc=GSE319132 | NCBI Gene Expression Omnibus, GSE319132 |

The following previously published dataset was used:

| Author(s) | Year | Dataset title | Dataset URL | Database and Identifier |
| --- | --- | --- | --- | --- |
| Fuping Q, Guanyun W, Gangcai X, Dong L | 2023 | Single-cell RNA-sequencing (scRNA-seq) of GFP + cells isolated from Tg(Brn3c:mGFP) 6 days zebrafish with GFP expression in all hair cells | https://www.ncbi.nlm.nih.gov/geo/query/acc.cgi?acc=GSE221471 | NCBI Gene Expression Omnibus, GSE221471 |

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
