## [Editor Report · eLife Assessment]

This study provides **valuable** data on the role of Hsd17b7, a gene involved in cholesterol biosynthesis, as a potential regulator of mechanosensory hair cell function. The authors used both zebrafish and the HEI cell line to examine the effects of deletion of Hsd17b7 on hair cell function and survival. While the study presents **convincing** evidence, the effect sizes observed across several experiments, including functional readouts such as the acoustic startle response, are modest, which raises questions about the biological significance of the proposed mechanism.

---

## [Referee Report · Reviewer #1 (Public review)]

Summary:

This study identifies HSD17B7 as a cholesterol biosynthesis gene enriched in sensory hair cells, with demonstrated importance for auditory behavior and potential involvement in mechanotransduction. Using zebrafish knockdown and rescue experiments, the authors show that loss of hsd17b7 reduces cholesterol levels and impairs hearing behavior. They also report a heterozygous nonsense variant in a patient with hearing loss. The gene mutation has a complex and somewhat inconsistent phenotype, appearing to mislocalize, reduce mRNA and protein levels, and alter cholesterol distribution, supporting HSD17B7 as a potential deafness gene.

The study presents an interesting deafness candidate with a complex mechanism, and highlights an underexplored role for cholesterol (and lipids) in hair cell function.

The authors were very responsive to the initial reviews, and the manuscript is now significantly stronger.

Strengths:

- HSD17B7 is a new candidate deafness gene with plausible biological relevance.

- Cross-species RNAseq convincingly shows hair-cell enrichment.

- Lipid metabolism, particularly cholesterol homeostasis, is an emerging area of interest in auditory function.

- The connection between cholesterol levels and MET is potentially impactful and, if substantiated, would represent a significant advance.

- The localization of HSD17B7 is reasonably convincing, despite the lack of a KO control: In HEI-OC1 cells, HSD17B7 localizes to the ER, as expected. In mouse hair cells, the staining pattern is cytosolic. The developmental increase between P1 and P4, and the higher expression in OHCs aligns nicely with RNAseq data.

Weaknesses:

- The pathogenic mechanism of the E182STOP variant is unclear: The mutant protein presumably does not affect WT protein localization, arguing against a dominant-negative effect. Yet, overexpression of HSD17B7-E182* alone causes toxicity in zebrafish and it binds and mislocalizes cholesterol in HEI-OC1 cells, suggesting some gain-of-function or toxic effect. In addition, the mRNA of the variant has low expression level, suggesting nonsense-mediated decay. The mechanistic conclusions of the study are therefore not as clear cut as one would had hoped, but it might just be a reflection of real biological complexity.

- The link to human deafness is based on a single heterozygous patient with no syndromic features. Given that nearly all known cholesterol metabolism disorders are syndromic, this raises concerns about causality or specificity. HSD17B7 is therefore, at this point, a candidate deafness gene, and not a fully established "novel deafness gene". This is acknowledged by the authors.

- This study does not directly investigate how reduced cholesterol levels affect MET. However, this is not a significant limitation given the study's scope, and it is reasonable that such detailed functional analyses are left to specialists in hair cell physiology.

---

## [Referee Report · Reviewer #2 (Public review)]

A summary of what the authors were trying to achieve.

The authors aim to determine whether the gene Hsb17b7 is essential for hair cell function and, if so, to elucidate the underlying mechanism, specifically the HSB17B7 metabolic role in cholesterol biogenesis. They use animal, tissue, or data from zebrafish, mouse, and human patients.

Strengths:

(1) This is the first study of Hsb17b7 in the zebrafish (a previous report identified this gene as a hair cell marker in the mouse utricle).

(2) The authors demonstrate that Hsb17b7 is expressed in hair cells of zebrafish and the mouse cochlea.

(3) In zebrafish larvae, a likely KO of the Hsb17b7 gene causes a mild phenotype in an acoustic/vibrational assay, which also involves a motor response.

(4) In zebrafish larvae, a likely KO of the Hsb17b7 gene causes a mild reduction in lateral line neuromast hair cell number and a mild decrease in the overall mechanotransduction activity of hair cells, assayed with a fluorescent dye entering the mechanotransduction channels.

(5) When HSB17B7 is overexpressed in a cell line, it goes to the ER, and an increase in Cholesterol cytoplasmic puncta is detected. Instead, when a truncated version of HSB17B7 is overexpressed, HSB17B7 forms aggregates that co-localize with cholesterol.

(6) It seems that the level of cholesterol in crista and neuromast hair cells decreases when Hsb17b7 is defective

Comments on the revised version:

Overall, the paper has been improved, but it still needs to be moderated regarding the observed effects and their qualification. I suggest expressing each effect as % {plus minus} SD and indicating it in the main text to inform the reader.

- The title " HSD17B7 is required for the function of sensory hair cells by regulating cholesterol Synthesis" should be moderated: "affects" instead of "required" would be better.

- In the abstract "conserved and essential role for HSD17B7-mediated cholesterol biosynthesis", the term essential seems overstated and premature

- In the discussion: "Collectively, these results suggest that the heterozygous c.544G>T (p.E182*) variant contributes to auditory dysfunction through potential pathogenic mechanisms: haploinsufficiency caused by reduced"...; "could contribute" would be safer.

- In the discussion: "In summary, our study identifies HSD17B7 as a critical regulator of cholesterol synthesis in sensory hair cells and as an essential factor in normal MET and sound-evoked sensory responses. "This part is still an overstatement. The effect in zebrafish is not directly shown to affect hearing, and startle reflex impairment is mild. It is not essential.

---

## [Author Response]

The following is the authors’ response to the original reviews.

**Public Reviews:**

**Reviewer #1 (Public review):**
(1) The pathogenic mechanism of the E182STOP variant is unclear. The mutant protein does not appear to affect WT protein localization, arguing against a dominant-negative effect. Yet, overexpression of HSD17B7-E182* alone causes toxicity in zebrafish and mislocalizes cholesterol in HEI-OC1 cells, suggesting a gain-of-function or toxic effect. In addition, the variant mRNA is expressed at a low level, consistent with nonsense-mediated decay. This apparent complexity and inconsistency need clearer explanation.

We appreciate the reviewer’s careful evaluation of this mechanistic complexity. Based on our combined molecular, cellular, and in vivo data, we propose that the pathogenic effect of the HSD17B7-E182* variant reflects a composite mechanism, rather than a classical dominant-negative effect.

At the transcript level, the E182* variant introduces a premature termination codon and shows markedly reduced mRNA abundance, consistent with partial degradation by nonsense-mediated mRNA decay. This reduction is expected to decrease overall HSD17B7 dosage, contributing a loss-of-function component. Unlike HSD17B7, the truncated HSD17B7^E182*^ mislocalizes cholesterol in HEI-OC1 cells, and overexpression alone reduces hair cell MET function and startle response in zebrafish embryos. We therefore propose that the truncated protein disturbing local cholesterol homeostasis, thereby exerts a toxic or ectopic gain-of-function.

We have revised the manuscript to clarify the dual-mechanism model.

(2) The link to human deafness is based on a single heterozygous patient with no syndromic features. Given that nearly all known cholesterol metabolism disorders are syndromic, this raises concerns about causality or specificity. The term "novel deafness gene" is premature without additional cases or segregation data.

We thank the reviewer for this important point. We fully agree that, based on a single heterozygous case without segregation data, it is premature to designate HSD17B7 as a novel deafness gene. Therefore, we have revised the manuscript to use the description of "candidate deafness genes".

(3) The localization of HSD17B7 should be clarified better: In HEI-OC1 cells, HSD17B7 localizes to the ER, as expected. In mouse hair cells, the staining pattern is cytosolic and almost perfectly overlaps with the hair cell marker used, Myo7a. This needs to be discussed. Without KO tissue, HSD17B7 antibody specificity remains uncertain.

We thank the reviewer for the constructive comments regarding HSD17B7 localization and antibody specificity.

Regarding subcellular localization, the original Figure 1K was intended to demonstrate the expression of HSD17B7 in mouse hair cells. To address this concern, we performed additional immunostaining on dissected organ of Corti sections at P1, P4, and P7 using higher magnification. Using parvalbumin as a hair cell marker, HSD17B7 displayed a partially punctate intracellular pattern in hair cells (revised Figure 1K). This pattern is consistent with localization to membrane-associated compartments, including the endoplasmic reticulum, and agrees with the ER-associated localization observed in HEI-OC1 cells and zebrafish hair cells. In mature hair cells, ER-associated signals may appear cytosolic and overlap with general hair cell markers such as Myo7a.

Regarding antibody specificity, although HSD17B7 knockout tissue was not available, we performed complementary validation experiments in HEI-OC1 cells. Cells were transfected with pCMV-Flag, pCMV-Flag-hHSD17B7WT, or pCMV-hHSD17B7WT-EGFP constructs and stained with anti-Flag, anti-EGFP, and anti-HSD17B7 antibodies. The HSD17B7 antibody signal showed strong co-localization with both FLAG- and EGFP-tagged HSD17B7 (revised Figure S1A and B), supporting its specificity.

**Reviewer #2 (Public review):**
(1) The statement that HSD17B7 is "highly" expressed in sensory hair cells in mice and zebrafish seems incorrect for zebrafish:(a) The data do not support the notion that HSB17B7 is "highly expressed" in zebrafish. Compared to other genes (TMC1, TMIE, and others), the HSB17B7 level of expression in neuromast hair cells is low (Figure 1F), and by extension (Figure 1C), also in all hair cells. This interpretation is in line with the weak detection of an mRNA signal by ISH (Figure 1G I"). On this note, the staining reported in I" does not seem to label the cytoplasm of neuromast hair cells. An antisense probe control, along with a positive control (such as TMC1 or another), is necessary to interpret the ISH signal in the neuromast.

We thank the reviewer for this detailed evaluation and agree that the description of HSD17B7 expression in zebrafish hair cells requires clarification.

To address this, we performed a quantitative comparison of average expression levels within neuromast hair cells using log-normalized single-cell RNA-seq data. This analysis shows that *hsd17b7* is expressed at a level comparable to several known MET-associated genes (e.g., *tmc1* and *lhfpl5a*) (revised Figure 1D). Regarding the pseudotime heatmap (Figure 1F), we now state that this analysis illustrates temporal expression dynamics within neuromast hair cell development.

In addition, we have clarified the interpretation of the whole-mount in situ hybridization data by emphasizing that the signal indicates spatial enrichment rather than high transcript abundance.

We have updated the figure panels, legends, and corresponding text in the Results section to reflect these changes.

(b) However, this is correct for mouse cochlear hair cells, based on single-cell RNA-seq published databases and immunostaining performed in the study. However, the specificity of the anti-HSD17B7 antibody used in the study (in immunostaining and western blot) is not demonstrated. Additionally, it stains some supporting cells or nerve terminals. Was that expression expected?

To assess antibody specificity, we performed validation experiments using distinct epitopes. In HEI-OC1 cells transfected with pCMV-Flag-HSD17B7, or pCMV-HSD17B7-EGFP constructs, immunostaining with anti-HSD17B7 showed strong co-localization with both FLAG- and EGFP-tag (revised Figure S1B). In addition, western blot analyses using the same constructs confirmed the specific detection of HSD17B7 protein (revised Figure S1B). These validation data have now been included as supplementary figures in the revised manuscript and provide independent supporting evidence for the specificity of the anti-HSD17B7 antibody.

(2) A previous report showed that HSD17B7 is expressed in mouse vestibular hair cells by single-cell RNAseq and immunostaining in mice, but it is not cited: Spatiotemporal dynamics of inner ear sensory and non-sensory cells revealed by single-cell transcriptomics. Jan TA, Eltawil Y, Ling AH, Chen L, Ellwanger DC, Heller S, Cheng AG. Cell Rep. 2021 Jul 13;36(2):109358. doi: 10.1016/j.celrep.2021.109358.

We have now cited this reference in the revised manuscript.

(3) Overexpressed HSD17B7-EGFP C-terminal fusion in zebrafish hair cells shows a punctiform signal in the soma but apparently does not stain the hair bundles. One limitation is the consequence of the C-terminal EGFP fusion to HSD17B7 on its function, which is not discussed.

We thank the reviewer for raising this important technical point. The apparent absence of an HSD17B7-EGFP signal in hair bundles is primarily due to the imaging strategy and the selection of representative images. In zebrafish hair cells, the EGFP signal within hair bundles is extremely strong. To better visualize the intracellular distribution of HSD17B7 within the hair cell soma, we selected representative confocal optical sections that were focused on the cell body rather than on the apical hair bundle plane. As a result, the hair bundle signal is not visible in the images shown.

Importantly, we agree that C-terminal EGFP fusion may potentially influence protein localization or function. We have therefore revised the Discussion to discuss this limitation and to clarify that our central conclusions regarding HSD17B7 function are primarily supported by loss-of-function analyses, rescue experiments using untagged mRNA, and cholesterol perturbation phenotypes, rather than relying solely on EGFP-tagged overexpression constructs.

(4) A mutant Zebrafish CRISPR was generated, leading to a truncation after the first 96 aa out of the 340 aa total. It is unclear why the gene editing was not done closer to the ATG. This allele may conserve some function, which is not discussed.

Targeting regions close to the ATG is indeed a commonly used strategy for CRISPR-mediated gene disruption. In this study, sgRNA selection was guided by online CRISPR design tools (CRISPRscan), prioritizing predicted cutting efficiency and specificity. This strategy resulted in a frameshift mutation introducing a premature stop codon after amino acid 96 of the 340-aa Hsd17b7 protein.

Importantly, this truncation removes most of the conserved catalytic core required for 17β-hydroxysteroid dehydrogenase activity, including key motifs involved in NAD(P)-binding and substrate recognition. Therefore, although the mutation does not occur immediately adjacent to the ATG, the resulting allele is predicted to lack enzymatic function. We have clarified this rationale and discussed the functional consequences of the truncation in the revised manuscript.

(5) The hsd17b7 mutant allele has a slightly reduced number of genetically labeled hair cells (quantified as a 16% reduction, estimated at 1-2 HC of the 9 HC present per neuromast). On a note, it is unclear what criteria were used to select HC in the picture. Some Brn3C:mGFP positive cells are apparently not included in the quantifications (Figure 2F, Figure 5A).

Upon re-evaluation, we recognized that the original figure annotations were not sufficiently clear and may have led to confusion regarding hair cell selection. In the original images, the absence of dashed outlines around some Brn3c:mGFP^+^ cells may have been misinterpreted as their exclusion from analysis. To address this issue, we have revised Figures 2F and 5A by updating the annotations to ensure that all Brn3c:mGFP^+^ hair cells within each neuromast are clearly visible and unambiguously included (revised Figures 2F and 6A). Corresponding figure legends have also been revised to clarify the criteria used for hair cell identification and quantification.

(6) The authors used FM4-64 staining to evaluate the hair cell mechanotransduction activity indirectly. They found a 40% reduction in labeling intensity in the HCs of the lateral line neuromast. Because the reduction of hair cell number (16%) is inferior to the reduction of FM4-64 staining, the authors argue that it indicates that the defect is primarily affecting the mechanotransduction function rather than the number of HCs. This argument is insufficient. Indeed, a scenario could be that some HC cells died and have been eliminated, while others are also engaged in this path and no longer perform the MET function. The numbers would then match. If single-cell staining can be resolved, one could determine the FM4-64 intensity per cell. It would also be informative to evaluate the potential occurrence of cell death in this mutant. On another note, the current quantification of the FM4-64 fluorescence intensity and its normalization are not described in the methods. More importantly, an independent and more direct experimental assay is needed to confirm this point. For example, using a GCaMP6-T2A-RFP allele for Ca2+ imaging and signal normalization.

We have revised the FM4-64 quantification strategy. Instead of measuring fluorescence intensity at the neuromast level, FM4-64 uptake was re-quantified at the single hair cell level. Hair cells within each neuromast were identified based on mGFP labeling, and the mean FM4-64 fluorescence intensity was measured for each individual hair cell. The average FM4-64 intensity per hair cell was then calculated for each neuromast and used for group comparisons (revised Figures 2F, 6B, and 8B, Figure S5B). The updated quantification method, normalization procedure, and analysis pipeline have now been described in the revised Methods section.

As supportive evidence, we further analyzed single-cell RNA-seq data from control and *hsd17b7* mutant hair cells (revised Figure 3). This analysis revealed dysregulation of multiple genes involved in the MET machinery, including reduced expression of tip-link–associated components and altered expression of other MET-related genes. While these transcriptional changes do not constitute a direct functional assay, they are consistent with perturbation of MET-associated pathways and complement the FM4-64 findings.

(7) The authors used an acoustic startle response to elicit a behavioral response from the larvae and evaluate the "auditory response". They found a significative decrease in the response (movement trajectory, swimming velocity, distance) in the hsd17b7 mutant. The authors conclude that this gene is crucial for the "auditory function in zebrafish".This is an overstatement:(a) First, this test is adequate as a screening tool to identify animals that have lost completely the behavioral response to this acoustic and vibrational stimulation, which also involves a motor response. However, additional tests are required to confirm an auditory origin of the defect, such as Auditory Evoked Potential recordings, or for the vestibular function, the Vestibulo-Ocular Reflex.

We thank the reviewer for highlighting the limitations in interpreting the acoustic startle assay. We have revised the manuscript to avoid overstatement and now describe the observed phenotype as a reduction in the behavioral response to acoustic and vibrational stimulation, rather than concluding a specific impairment of auditory function.

(b) Secondly, the behavioral defects observed in the mutant compared to the control are significantly different, but the differences are slight, contained within the Standard Deviation (20% for velocity, 25% for distance). To this point, the Figure 2 B and C plots are misleading because their y-axis do not start at 0.

We have corrected Figures 2B and 2C so that the y-axes start at zero, thereby providing a more transparent visualization of the behavioral differences. The figure legends have also been revised to clarify the presentation of the data.

(8) Overexpression of HSD17B7 in cell line HEI-OC1 apparently "significantly increases" the intensity of cholesterol-related signal using a genetically encoded fluorescent sensor (D4H-mCherry). However, the description of this quantification (per cell or per surface area) and the normalization of the fluorescent signal are not provided.

The quantification of the D4H-mCherry signal in HEI-OC1 cells was performed at the single-cell level. Specifically, individual cells were segmented based on morphology, and the mean fluorescence intensity of D4H-mCherry per cell was measured. To account for variability in cell size and imaging conditions, fluorescence intensity was normalized to the background signal measured from cell-free regions in the same field of view. We have now clarified the quantification strategy and normalization procedure in the revised Methods and Results sections.

(9) When this experiment is conducted in vivo in zebrafish, a reduction in the "DH4 relative intensity" is detected (same issue with the absence of a detailed method description). However, as the difference is smaller than the standard deviation, this raises questions about the biological relevance of this result.

We have now clarified the quantification strategy and normalization procedure in the revised Methods and Results sections.

(10) The authors identified a deaf child as a carrier of a nonsense mutation in HSB17B7, which is predicted to terminate the HSB17B7 protein before the transmembrane domain. However, as no genetic linkage is possible, the causality is not demonstrated.

We thank the reviewer for raising this important point. Unfortunately, we were unable to obtain the parents' genetic testing data to perform formal genetic and linkage analysis. To address this limitation, we have revised the manuscript to avoid causal overstatement and now describe the HSD17B7 E182* variant as a candidate pathogenic variant associated with hearing loss. Importantly, our functional analyses in zebrafish and cell-based systems demonstrate that the E182* truncation abolishes key biological activities of HSD17B7, including subcellular localization, cholesterol regulation, mechanotransduction-related activity, and behavioral responses. These convergent functional data provide biological support for the potential pathogenic relevance of this variant.

(11) Previous results obtained from mouse HSD17B7-KO (citation below) are not described in sufficient detail. This is critical because, in this paper, the mouse loss-of-function of HSD17B7 is embryonically lethal, whereas no apparent phenotype was reported in heterozygotes, which are viable and fertile. Therefore, it seems unlikely that heterozygous mice exhibit hearing loss or vestibular defects; however, it would be essential to verify this to support the notion that the truncated allele found in one patient is causal.Hydroxysteroid (17beta) dehydrogenase 7 activity is essential for fetal de novo cholesterol synthesis and for neuroectodermal survival and cardiovascular differentiation in early mouse embryos.Jokela H, Rantakari P, Lamminen T, Strauss L, Ola R, Mutka AL, Gylling H, Miettinen T,Pakarinen P, Sainio K, Poutanen M. Endocrinology. 2010 Apr;151(4):1884-92. doi: 10.1210/en.2009-0928. Epub 2010 Feb 25.

We thank the reviewer for raising this important point. We acknowledge that previous work has shown that complete loss of Hsd17b7 in mice is embryonically lethal, whereas heterozygous animals are viable and fertile (Jokela et al., 2010). Notably, this study primarily focused on embryonic development, cholesterol metabolism, and cardiovascular and neuroectodermal survival, and auditory or vestibular functions were not specifically examined. Therefore, subtle or sensory organ–specific phenotypes in heterozygous mice cannot be excluded.

The human variant identified in this study (E182*) is a nonsense mutation predicted to truncate the HSD17B7 protein prior to the transmembrane and cytoplasmic domains. We therefore present it as a candidate loss-of-function variant, providing supportive human genetic evidence that is consistent with our functional analyses in zebrafish hair cells, rather than as definitive proof of causality. We have revised the manuscript to clarify these points and to acknowledge this limitation.

(12) The authors used this truncated protein in their startle response and FM4-64 assays. First, they show that contrary to the WT version, this truncated form cannot rescue their phenotypes when overexpressed. Secondly, they tested whether this truncated protein could recapitulate the startle reflex and FM4-64 phenotypes of the mutant allele. At the homozygous level (not mentioned by the way), it can apparently do so to a lesser degree than the previous mutant. Again, the differences are within the Standard Deviation of the averages. The authors conclude that this mutation found in humans has a "negative effect" on hearing, which is again not supported by the data.

We thank the reviewer for this important comment. We agree that the overexpression strategy employed in this study does not fully replicate the endogenous heterozygous state observed in patients, and that the magnitude of the observed effects varies across samples. Accordingly, our experiments were not intended to demonstrate a definitive causal role of the HSD17B7 ^E182*^ variant in hearing loss.

Instead, the overexpression assays were designed to assess whether the truncated HSD17B7 protein displays abnormal cellular properties and whether its presence can interfere with processes relevant to hair cell function. Under these conditions, HSD17B7^E182*^ exhibited aberrant subcellular localization, altered intracellular cholesterol distribution, and was associated with reduced FM4-64 uptake and changes in startle-associated behaviors, whereas the wild-type protein did not.

We revised the manuscript to moderate our conclusions. Rather than claim that the E182* mutation has a definitive “negative effect on auditory function,” we now describe it as a functionally compromised allele that disrupts cholesterol distribution and MET-related activity under overexpression conditions, providing mechanistic support consistent with our zebrafish loss-of-function data and the identification of this variant in a patient with hearing loss. In addition, the "negative effect" statement was based on the result that overexpression of the E182* mutation in wild-type embryos caused the compromised MET function and startle response defect.

(13) The authors looked at the distribution of the HSB17B7 in a cell line. The WT version goes to the ER, while the truncated one forms aggregates. An interesting experiment consisted of co-expressing both constructs (Figure S6) to see whether the truncated version would mislocalize the WT version, which could be a mechanism for a dominant phenotype. However, this is not the case.

We thank the reviewer for raising this important point regarding a potential dominant-negative mechanism. Consistent with the reviewer’s interpretation, we found that HSD17B7^WT^ predominantly localizes to the endoplasmic reticulum, whereas the truncated HSD17B7^E182*^ protein forms intracellular aggregates. Importantly, we further observed that the E182* mutation markedly reduces the stability of both HSD17B7 mRNA and protein, resulting in substantially decreased abundance of the truncated protein (Figure S6B–E). As a consequence, the cellular levels of HSD17B7^E182* are abnormally low.

Based on these findings, we consider it unlikely that the E182* variant exerts its effect through interference with the wild-type protein. Our results suggest that the heterozygous c.544G>T (p.E182*) variant contributes to auditory dysfunction through potential pathogenic mechanisms: 1, haploinsufficiency caused by reduced HSD17B7 expression, 2, functional impairment due to altered protein subcellular localization and cholesterol distribution.

We have revised the Results and Discussion sections. Our conclusions now emphasize that the functional impact of this variant is attributable to decreased effective HSD17B7 dosage, consistent with the observed defects in cholesterol synthesis, MET-related activity, and auditory-associated phenotypes in our model.

(14) Through mass spectrometry of HSB17B7 proteins in the cell line, they identified a protein involved in ER retention, RER1. By biochemistry and in a cell line, they show that truncated HSB17B7 prevents the interaction with RER1, which would explain the subcellular localization.

Consistent with the reviewer’s interpretation, wild-type HSD17B7 interacts with RER1, a protein known to participate in ER retention, whereas this interaction is lost in the truncated HSD17B7 variant. We propose that RER1 is an interacting partner of HSD17B7, providing a mechanistic explanation for the protein's subcellular localization.

(15) Information and specificity validation of the HSB17B7 antibody are not presented. It seems that it is the same used on mice by IF and on zebrafish by Western. If so, the antibody could be used on zebrafish by IF to localize the endogenous protein (not overexpression as done here). Secondly, the specificity of the antibody should be verified on the mutant allele. That would bring confidence that the staining on the mouse is likely specific.

We thank the reviewer for raising this important point regarding antibody specificity and validation. Information on the HSD17B7 antibody and its validation has been provided in our response to comment 1, where we described the use of antibodies recognizing different epitopes and the experimental strategies employed to assess specificity (revised Figure S1A and B).

Although the same antibody was used for Western blot analysis in zebrafish samples, its performance in immunofluorescence staining of zebrafish tissues was suboptimal, with relatively high background. For this reason, we did not rely on this antibody for endogenous Hsd17b7 localization in zebrafish by immunofluorescence and instead employed tagged constructs for subcellular localization analyses. This approach provides more reliable and interpretable localization information under the current experimental conditions.

**Recommendations for the authors:**

**Reviewing Editor Comments:**
Suggested revisions to help improve the study and the eLife Assessment:(1) FM4-64 uptake: Isolate the effect of hair cell loss and MET reduction.(2) Clarify the mechanistic model: Is the mutant protein pathogenic due to toxicity, lack of expression or function, or both? Come up with a clearer causal chain of events.(3) Mouse immunostaining: Validate the HSD17B7 antibody, and since mouse RNAseq data (gEAR database) suggest that HSD17B7 expression increases dramatically between P0-P5, show this developmental progression by immunostaining of the mouse organ of Corti at P0, P3, and P5.(4) The HSD17B7-E182* expression disrupts cholesterol (D4H staining) in OC1 cells. This should also be demonstrated in the mutant zebrafish.(5) Structural modeling of E182* is uninformative; half the protein is absent. This kind of analysis is better suited for missense variants. Suggest removing this analysis.

We thank the Reviewing Editor for these constructive suggestions. The major points raised here substantially overlap with the concerns raised in the public reviews. In response, we have:

(1) revised FM4-64 quantification and interpretation to better distinguish hair cell loss from MET impairment;

(2) Clarify the mechanistic mode. Mechanistically, the mutation decreases mRNA abundance and significantly reduces protein levels. Moreover, expression of the p.E182* mutation disrupted the interaction between HSD17B7 and the ER retention receptor RER1, leading to aberrant subcellular localization and altered cholesterol distribution, thereby exacerbating HC dysfunction.

(3) provided additional validation of the HSD17B7 antibody using antibodies targeting distinct epitopes, and extended mouse organ of Corti immunostaining to postnatal stages P1, P4, and P7 to demonstrate the developmental upregulation of HSD17B7 expression;

(4) added in vivo zebrafish experiments demonstrating that expression of HSD17B7^E182*^ disrupts cholesterol distribution in hair cells, consistent with the effects observed in HEI-OC1 cells using D4H staining;

(5) removed the structural modeling of the E182* variant.

**Recommendations for the authors:**

The recommendations from Reviewer #1 and Reviewer #2 were carefully considered and addressed. Most of these points overlap with the public reviews and the Reviewing Editor's comments and have been addressed through a revised mechanistic interpretation, additional clarifications in the Methods, more moderate claims regarding auditory function and human genetics, and the removal or revision of potentially misleading analyses. In addition, a number of minor issues were corrected, including missing or incorrect references, repetitive or unclear statements in the Introduction, insufficient methodological details, imprecise terminology, and typographical or formatting errors. Collectively, these revisions improve the clarity, rigor, and transparency of the study without altering its central conclusions.